# Grounding zone subglacial properties from calibrated active source seismic methods

Huw J. Horgan[1], Laurine van Haastrecht[1], Richard B. Alley[2], Sridhar Anandakrishnan[2], Lucas Beem[3], Knut Christianson[4], Atsuhiro Muto[5], and Matthew Siegfried[6]

[1]Antarctic Research Centre, Victoria University of Wellington, Wellington, New Zealand
[2]Department of Geosciences and Earth and Environmental Systems Institute, Pennsylvania State University, University Park, Pennsylvania 16802, USA
[3]Department of Earth Sciences, Montana State University, Bozeman, Montana 59717, USA
[4]Earth and Space Sciences, University of Washington, Seattle, Washington 98195, US
[5]Department of Earth and Environmental Science, Temple University, Philadelphia, Pennsylvania 19122, USA
[6]Department of Geophysics, Colorado School of Mines, Golden, Colorado 80401, USA

**Correspondence:** Huw Horgan (huw.horgan@vuw.ac.nz)

**Abstract.** The grounding zone of Whillans Ice Stream, West Antarctica, exhibits an abrupt transition in basal properties from the grounded ice to the ocean cavity over distances of less than 0.5–1 km. Active source seismic methods reveal the downglacier-most grounded portion of the ice stream is underlain by a relatively stiff substrate (relatively high shear wave velocities of $1100 \pm 430$ m s$^{-1}$) compared to the deformable till found elsewhere beneath the ice stream. Changes in basal reflectivity in our study area cannot be explained by the stage of the tide. Several kilometers upstream of the grounding zone, layers of subglacial water are detected, as are regions that appear to be water layers but are less than the thickness resolvable by our technique. The presence of stiff subglacial sediment and thin water layers upstream of the grounding zone supports previous studies that have proposed the dewatering of sediment within the grounding zone and the trapping of subglacial water upstream of the ocean cavity. The setting enables calibration of our methodology using returns from the floating ice shelf. This allows a comparison of different techniques used to estimate the sizes of the seismic sources, a constraint essential for the accurate recovery of subglacial properties. We find a strong correlation (coefficient of determination=0.46) between our calibrated method and a commonly used multiple bounce method, but our results also highlight the incomplete knowledge of other factors affecting the amplitude of seismic sources and reflections in the cryosphere.

## 1   Introduction

Grounding zones mark the transition from grounded to floating ice, standing sentinel over much of the contribution of glaciers and ice sheets to sea level. Within the grounding zone the location where the ice sheet ceases contact with the bed (the grounding line) is primarily determined by ice thickness, bed elevation, and the stage of the tide. In the Antarctic, tidally induced migration

of the grounding line within the grounding zone varies from near zero in the case of abrupt changes in bed elevation and/or ice thickness, to up to 10 kilometers in the case of gently sloping ice plains (Brunt et al., 2011; Dawson and Bamber, 2020). Along with grounding line migration, tides correlate with ice velocity changes upstream and downstream of the grounding zone. Observations include daily velocity variability on Bindschadler Ice Stream (Anandakrishnan et al., 2003), twice daily stick-slip displacement on Whillans Ice Plain (Bindschadler et al., 2003; Winberry et al., 2009; Walter et al., 2011), daily and spring–neap velocity variability on the Ronne–Filchner Ice Shelf, Ross Ice Shelf and Byrd Glacier (Rosier and Gudmundsson, 2020; Brunt et al., 2011; Marsh et al., 2013) and spring-neap tidal velocity variability on Rutford Ice Stream (Gudmundsson, 2007). Observed velocity variability has generally been attributed to tidal changes in the force balance interacting with the underlying till rheology (Bindschadler et al., 2003; Gudmundsson, 2007; Winberry et al., 2009). Subsequent studies have attributed Rutford Ice Stream's spring-neap velocity variability to changes in subglacial pore water pressure (Rosier et al., 2015), while on Rutford and elsewhere others have pointed to contact with ice shelf pinning points and at the grounding zone as the causes of observed velocity changes (Robel et al., 2017; Minchew et al., 2017; Rosier and Gudmundsson, 2020).

Early efforts to model tidal deflection of ice shelves primarily addressed vertical displacement and the associated development of strand cracks and basal crevasssing at the grounding zone (Holdsworth, 1969, 1977). These models, termed stiff-bed fixed grounding line models by Sayag and Worster (2013), do not allow the grounding line to migrate, nor do they allow the underlying bed to deform. Despite inconsistencies in the retrieved elastic properties, subsequent applications of these models have successfully reproduced surface displacement (e.g. Vaughan, 1995; Schmeltz et al., 2002) with models accounting for basal crevassing (Rosier et al., 2017) and treating the ice as a viscoelastic material (Wild et al., 2017) shown to be more consistent with observations. The importance of grounding line migration for ice dynamics and the sensitivity of ice flow to tidal forcing has prompted renewed examination of the effect of tides on grounding line migration distances and subglacial conditions both within and upstream of the grounding zone. Sayag and Worster (2011) combined laboratory observations and an elastic sheet model in an analysis that allowed the grounding line to migrate over an elastic bed. Their approach was extended to the implications for subglacial water pressure (Sayag and Worster, 2013), showing pressure gradients alternating direction upstream of the grounding zone forming migrating barriers to subglacial water flow. Walker et al. (2013) used a fixed grounding line model with no vertical displacement at the grounding line and a viscoelastic ice sheet–shelf overlying an elastic bed. This approach resulted in alternating pressure gradients that may act to draw water from the ocean cavity at low tide and force it upstream at high tide. Tsai and Gudmundsson (2015) applied a novel elastic fracture approach to grounding line migration, which resulted is migration distances significantly different to elastic beam or hydrostatic approaches. Notably, Tsai and Gudmundsson (2015) demonstrated an asymmetry in grounding line migration whereby for a constant surface slope and a constant coastward bed slope, the grounding line migrates upstream as the tide rises from mean sea level much further than it propagates downstream when the tide falls from mean sea level. The subglacial system can also filter forcings leading to velocity changes that occur at unexpected frequencies (e.g. Rosier et al., 2015). Robel et al. (2017) attributed such behavior to the visoelastic response of the ice shelf as it responds to changes in contact and buttressing at the grounding zone and pinning points. Alternatively, Warburton et al. (2020) coupled processes of upstream fluid flow beneath an elastic sheet and drainage through porous till and

showed ice streams and ice shelves can respond at a range of frequencies and also suggested ocean water may be retained in the subglacial system depending on the porosity of the till.

Grounding zones have been directly observed in only a few locations around Antarctica. Beneath Langovde Glacier in East Antarctica Sugiyama et al. (2014) reported a substrate of fine sediment with decimeter scale dropstones, along with an
incursion of sea water far beyond the previously mapped grounding line. In the ocean cavity proximal to the grounding line of McKay Glacier, Powell et al. (1996) imaged a diverse range of glaciomarine lithologies, ranging from soft till to bedrock and dropstone boulders. Approximately 3 km downstream from Whillans Ice Stream's grounding zone, the WISSARD program (Fricker et al., 2010) observed an ice shelf melt-out deposit with a mixture of soft mud and rock clasts (Scherer et al., 2015). Begeman et al. (2018) reported oceanographic and geophysical observations from the WISSARD borehole where they found a
highly stratified water column with basal melt rates of less than 0.1 m a$^{-1}$. To further investigate the basal properties beneath Whillans Ice Stream's grounding zone we here revisit the active source seismic data reported by Horgan et al. (2013b) and apply and extend amplitude analysis methods previously used in studies addressing the basal boundary of glaciers and ice sheets (e.g. Anandakrishnan, 2003b; Smith, 2007; Holland and Anandakrishnan, 2009; Brisbourne et al., 2017; Zechmann et al., 2018; Muto et al., 2019). These methods require source amplitude and path effects to be estimated, which is often
challenging due to variability in source and receiver coupling, and strong vertical gradients in density and seismic velocity in the firn. Acquiring data over the ocean cavity allows calibration of these methods due to the presence of a known ice–water reflection interface. This allows us to use and expand on the methods of Holland and Anandakrishnan (2009) (hereafter referred to as H&A2009). H&A2009 reviewed active source seismic methods for the recovery of subglacial properties, outlined best practices for reducing uncertainties, and presented new strategies for source size determination. Our application and extension
of their methods enables a robust estimate of elastic properties beneath the ice at a relatively high spatial resolution. Our profile data cover approximately 50 line kilometers. The nominal horizontal resolution of our method is 240 m (based on the spatial footprint of a 100 Hz wave in a 3860 m s$^{-1}$ medium at a depth of 760 m) and we are able to image the top and bottom of a water layer $>=$ 3.6 m thick ($\lambda/4$, where $\lambda$ denotes wavelength, of a 100 Hz wave in a 1440 m s$^{-1}$ medium). In theory, water layers down to $\lambda/32$ (0.45 m) can be imaged, however amplitudes from these layers may not be representative of their elastic
properties (e.g. Booth et al., 2012). To explore the relationship between the tidal stage and our results, we also present the timing and tidal stage of our experiment, and Global Navigation Satellite System (GNSS) repeat transects along two profiles crossing the grounding zone.

## 2   Data and Methods

We performed amplitude analysis of data from four transects that cross the grounding zone of Whillans Ice Stream (Figure 1).
These data were acquired in the austral summer of 2011/2012. Acquisition was composed of an explosive seismic source detonated at approximately 27 m depth, with charge sizes of 0.4 kg (Line 1) and 0.8 kg (Lines 2, and 4) and 0.85 kg (Line 3) at a nominal shot spacing of 240 m. Each of Line 3's 0.85 kg charge was composed of one 0.4 kg charge and three narrower 0.15 kg charges. All other charges were composed of equal diameter 0.4 kg charges. The time between burial and detonation

varied but always exceeded 24 hours. Geophones were buried approximately 0.5 m beneath the snow surface at 20 m spacings, and consisted of alternating single-string 40 Hz geophones (even channels) and 5-element 40 Hz georods (odd channels, Voigt et al., 2013). Acquisition used an asymmetric split spread with near and far shot–receiver offsets of 10 m and 1430 m. Seismic imaging and grounding zone determination at Whillans Ice Stream is presented in Horgan et al. (2013b).

5     Following H&A2009, the amplitudes reflected off of the base of the ice and recorded at our geophones ($A_i$, where $i$ denotes the receiver index) are related to our source amplitude ($A_0$) by:

$$A_i = A_0 \gamma_i R(\theta) e^{-\alpha s_i} \quad \text{(Equation 1, H\&A2009)}, \tag{1}$$

where $R(\theta)$ denotes the angle ($\theta$) dependent reflection coefficient at the base of the ice described by the Zoeppritz equations (e.g. Aki and Richards, 1980). During travel along the path length ($s_i$) from the source to the receiver, amplitudes are modified 10   by path effects ($\gamma_i$) and attenuation ($\alpha$), all of which are discussed below. Both $A_0$ and $\gamma_i$ are amplitudes relative to a reference range (typically $d_0 = 1$ m, Holland and Anandakrishnan, 2009; Shearer, 2009).

## 2.1   Seismic Velocity Model

Tracing seismic ray paths between the source and receivers requires knowledge of the firn and ice column's seismic velocity. To achieve this we estimated a one-dimensional (1D) velocity model using shallow seismic-refraction techniques. During shallow 15   refraction surveying a hammer source was recorded at 0.5 m horizontal intervals with near and far offsets of 0.5 m and 579 m. A velocity model (Figure 2) was then calculated using first-break arrival times and the $\tau$-$p$ (intercept time–slowness) method (e.g. Shearer, 2009), which assumes that the velocity monotonically increases with depth. This method estimated a velocity of 3840 m s$^{-1}$ at 80 m depth. Below this depth our velocity model consists of an extrapolation to a $V_p$ corresponding to -20°C (3860 m s$^{-1}$; Kohnen, 1974) which is kept constant to the ice base. Kohnen (1974) demonstrated a decrease in $V_p$ of 2.3 m s$^{-1}$ 20   per degree C decrease in temperature, so our velocity is fairly insensitive to our choice of temperature. Also implicit in our use of a 1D velocity model is an assumption that seismic velocity does not vary laterally within the survey area.

## 2.2   Amplitude Picking

Amplitudes were picked on frequency-filtered and amplitude-scaled shot records guided by common depth point stacked profiles. On every shot record we attempted to digitize the direct arrival, primary bed return, and first long-path multiple of 25   the bed return (Figure 3). The low impedance-contrast at the ice-bed interface meant the long-path multiple could not be reliably picked in the grounded part of the profiles. Amplitude picking selected the zero crossing preceding the side-lobe of the wavelet. Amplitude extraction was then performed on shot records with only bandpass filtering applied. Amplitudes were extracted within the wavelet encompassing the first side lobe, the central lobe, and the next side lobe. Within this wavelet, peak positive, peak negative, and root mean squared (RMS) amplitudes were extracted. We avoided picking bed returns where direct 30   arrival energy interferes with the bed wavelet. Our data are from ice thicknesses of approximately 730–790 m and direct arrivals interfere with the reflection from the base of the ice beyond offsets of approximately 700 m. While the channels with 5-element georods showed better signal to noise ratios for imaging, we here present an analysis of the single-string geophones as their

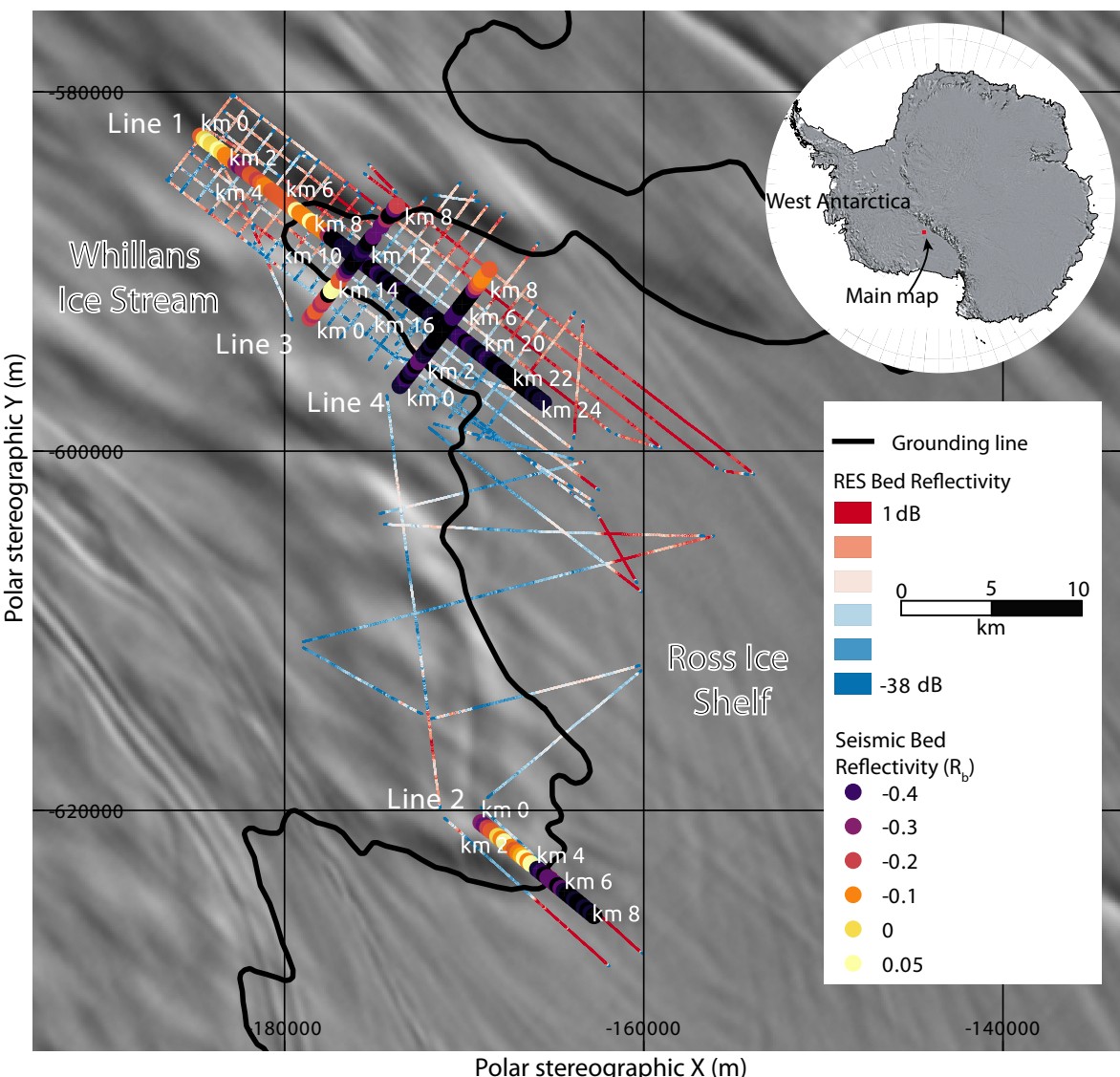

**Figure 1.** Location map showing the seismic profiles (Lines 1–4) crossing the grounding zone of Whillans Ice Stream. Radio echo sounding (RES) basal reflectivity from Christianson et al. (2016). Seismic bed reflectivity ($R_b$) from this study. Background imagery from MODIS MOA (Haran et al., 2005) and grounding line from Bindschadler et al. (2011). Polar stereographic projection (meters) with a true scale at 71° south.

amplitudes exhibit less channel to channel variability the cause of which we attribute to more variability in coupling when burying the georods. Our analysis also uses the RMS amplitudes, with the positive and negative peaks used to define polarity.

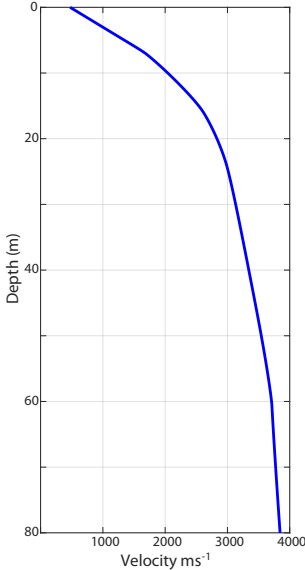

**Figure 2.** One dimensional compressional wave velocity profile estimated using the $\tau$-p method.

We tested the use of peak amplitudes and fixed wavelet length approaches and found both resulted in a greater distribution of source sizes, and less robust estimates of basal reflectivity.

## 2.3 Path effects

Path effects $(\gamma_i)$ modify the source amplitude during its propagation to the receiver. We calculated the total path effects as

$$\gamma_i = \frac{\cos\theta_i}{s_i}\sqrt{\frac{z_0}{z_1}} \tag{2}$$

where $\theta_i$ denotes the angle between the incoming ray and normal incidence, $z_0$, $z_1$ denote the acoustic impedance at the source and receiver respectively, and $s_i$ denotes the path length traveled between the source and receiver. Equation 2 therefore accounts for the angle at which the incoming ray arrives at the vertical-component receivers $(\cos\theta_i)$, amplitude scaling due to the different acoustic impedance at the source and receiver $(\sqrt{\frac{z_0}{z_1}}$, e.g. Shearer, 2009), and geometric spreading along the ray path $(1/s_i)$. We estimated all near-field effects using the 1D velocity model (Figure 2) and the density–compressional-wave velocity relationship of Kohnen and Bentley (1973). The high vertical gradients in density and velocity in polar firn lead to a $\cos\theta_i$ correction $\approx 1$, as $\theta_i \approx 0$, and a significant $\sqrt{\frac{z_0}{z_1}}$ correction $(\sim\sqrt{10})$ due to the different source and receiver burial depths.

H&A2009 noted that placing receivers at a free surface results in a further doubling of recorded amplitudes for normal incidence returns. We tested including free surface amplification but did not apply it to the analysis presented here due to the burial of our receivers, although the shallow burial depth of 0.5 m may justify its inclusion. If included, this additional amplification would have resulted in a halving of the source sizes for two of our methods (the multiple bounce method and

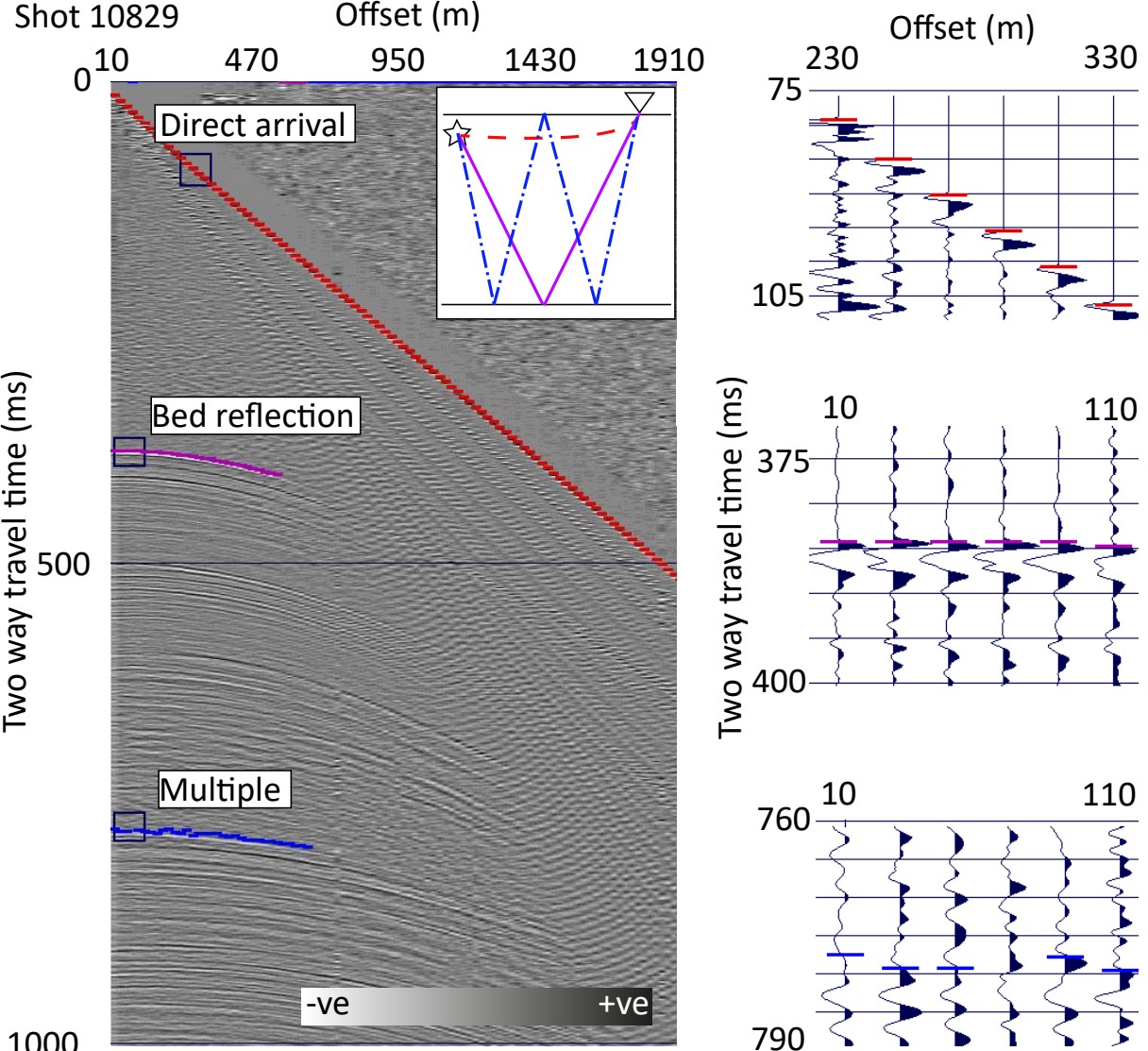

**Figure 3.** Left panel: example shot record from floating portion of Line 2 (kilometer 4.8-6.7). Left panel inset shows schematic travel paths for direct (red), primary (purple), and multiple (red) rays. Right hand panels show wavelets and picks for the direct arrival (top), primary return (middle), and multiple return (bottom).

the known reflector method (Sections 2.4.1 and 2.4.3, Table 1)). Including free surface amplification would have had a small effect ($<17\%$) on the direct path method source size median values (Section 2.4.2, Table 1). Regardless of whether free surface amplification is included or excluded, our choice of preferred method for estimating $A_0$ would not change. The recovered bed properties also would not change as the same path effects used to calculate source size are later used to estimate bed properties.

## 2.4 Source size and attenuation

Source size ($A_0$) is often estimated using the ratio of the primary bed return amplitude ($A_i$) and the long path multiple amplitude ($A_{m,i}$) (e.g. Röthlisberger, 1972; Smith, 1997; Peters et al., 2008; Brisbourne et al., 2017; Zechmann et al., 2018). This approach, termed the multiple bounce method by H&A2009, removes the need for an independent estimate of attenuation. However, low impedance contrast at the bed, low signal to noise ratios, or closely spaced subglacial reflectors, can all complicate the multiple bounce method of determining source amplitude. Here we explore this and other methods for determining the source amplitude because more-accurate source-amplitude estimates will enable improved investigation of the basal properties resolved by seismic surveys. These methods fall into three categories: (1) multiple bounce methods (2) direct arrival methods, and (3) known reflector methods. We present the results for each of the four profiles individually as three different source sizes and configurations were used.

### 2.4.1 Multiple bounce methods

Our multiple bounce methods used the primary–multiple amplitude ratio to estimate $A_0$ and followed H&A2009. The first method requires near-normal incidence returns but does not require knowledge of attenuation ($\alpha$):

$$A_{0,i} = \frac{A_i^2}{A_{m,i}} \frac{1}{2\gamma_i} \qquad \text{(Equation 6, H\&A2009)}, \tag{3}$$

and the second method requires close to normal incidence returns and an estimate of attenuation:

$$A_{0,i} = \frac{A_i^2}{A_{m,i}} \frac{\gamma_{m,i}}{\gamma_i^2} e^{\alpha(2d_i - d_{m,i})} \qquad \text{(Equation 7, H\&A2009)}, \tag{4}$$

where $d_i$ and $d_{m,i}$, and $\gamma_i$ and $\gamma_{m,i}$ denote the path length, and path amplitude factor (Equation 2) for the primary and multiple bed returns respectively. $A_0$ is then calculated as the average $A_{0,i}$ for each shot. Equations 3 and 4 give near identical $A_0$ estimates with root mean squared differences $\approx 0.1\%$. Henceforth for the amplitude ratio method we report only the results from Equation 4 with an angle cut off of $<10°$ and assuming an attenuation $\alpha = 0.27$ km$^{-1}$ (following Horgan et al., 2011). This attenuation corresponds to a seismic quality factor (Q) of 30–300 for 10–100 Hz waves in a 3860 m s$^{-1}$ medium. H&A2009 noted that Equation 4 is weakly dependent on uncertainties in $\alpha$. Long-path multiples from shots in which the primary reflections were from the interface between ice and seismically thick water resulted in 60, 19, 9, and 24 estimates of $A_0$ for Lines 1–4, respectively (left column Figure 4, $A_{0MB}$ columns Table 1).

### 2.4.2 Direct path methods

Two methods were used to estimate source amplitude from the direct arrival amplitudes ($B_i$). Direct arrivals have successfully been used to determine source size (Muto et al., 2019) and to normalise shot records (Brisbourne et al., 2017). Following H&A2009:

$$B_i = A_0 \gamma_{d,i} e^{-\alpha s_{d,i}} \qquad \text{(Equation 8, H\&A2009)}, \tag{5}$$

where $B_i$ denotes direct arrival amplitude at receiver index $i$, and $s_{d,i}$ and $\gamma_{d,i}$ are the direct arrival path lengths and path amplitude factors. We first estimated $A_0$ using the direct-path pair method of H&A2009 (H&A2009, Equation 9). This method uses receiver pairs where the ratio of path lengths $s_2/s_1 = 2$, and where the offset is sufficient that depth averaged attenuation can be assumed the same. This negates the need for an independent attenuation estimate. Our acquisition geometry did not result in pairs where $s_2/s_1 = 2$ exactly so an acceptance distance $(x_1)$ was set such that pairs were used if $s_2 >= 2s_1 - x_1 \wedge s_2 <= 2s_1 + x_1$. We set $x_1 = 14$ m through trial and error, looking for the minimum $x_1$ that resulted in multiple estimates of $A_0$ for all shots. This resulted in a mode of 8 pairs per shot (mean of 7.7, standard deviation of 3.7). $A_0$ direct pair estimates are shown in Figure 4 (centre left column) and Table 1 ($A_{0DP}$ columns).

We also investigated $A_0$ estimation using all direct arrival amplitudes by fitting the observed $B_i$ values to Equation 5 and minimizing the misfit to determine optimal $A_0$ and $\alpha$ values. We refer to this method as the direct path linear intercept method, because

$$\ln \frac{B_i}{\gamma_{d,i}} = -\alpha s_{d,i} + \ln A_0$$

shows that in $-s_i$ versus $\ln \frac{Bi}{\gamma_i}$ space every shot record should exhibit a common gradient $(\alpha)$, and independent y-intercepts representing $\ln A_0$. Despite this linear form we solved for best fitting parameters directly from Equation 5 using non-linear regression. We restricted our direct arrival analysis to returns from offsets greater than 450 m, and testing up to an offset limit of $> 800$ m did not result in significantly different $A_0$ and $\alpha$ estimates. For both direct path methods, path effects $(\gamma_{d,i})$ were estimated using both Equation 2 and by estimating wavefront energy using ray theory (Section 6.2 of Shearer, 2009, modified to account for different outgoing and incoming angles). The wavefront energy approach did not result in better $A_0$ estimates, with a larger distribution and poorer correlation with the known reflector method. We therefore present results using Equation 2, consistent with our other source size estimates. $A_0$ direct linear intercept estimates are shown in Figure 4 (centre right column) and Table 1 ($A_{0LI}$ columns).

### 2.4.3 Known reflector methods

Reflections from a known impedance contrast, in this case the floating ice shelf overlying the ocean cavity, allow another method of determining $A_0$. We estimated a best fitting $A_0$ for each ice shelf shot by non-linear regression of Equation 6 (Equation 10, H&A2009).

$$R(\theta) = \frac{A_i}{A_0} \frac{1}{\gamma_i} e^{-\alpha s_{d,i}} \qquad \text{(Equation 10, H\&A2009)}, \tag{6}$$

We determined the optimal $A_0$ for each floating shot by minimizing the root-mean-squared misfit between the reflection amplitudes resulting from the Zoeppritz equations for the seismic properties in Table 2, and the observed bed reflection amplitudes ($R(\theta)$, Equation 6). To account for the possibility that englacial debris may be present in the basal ice we also optimised the seismic properties of the ice used in the Zoeppritz equations while keeping the underlying water properties constant. We allowed the basal ice to vary within a range encompassing debris contents of 0–20% by volume. The range of seismic velocities for this basal ice was estimated using a Bruggeman mixing model following Röthlisberger (1972). We refer to this method as

**Table 1.** Source size ($A_0$) estimates.

| Line | Source Size (kg) | $A_{0MB}$ Median | $A_{0MB}$ Mean | $A_{0MB}$ Std | $A_{0DP}$ Median | $A_{0DP}$ Mean | $A_{0DP}$ Std | $A_{0LI}$ Median | $A_{0LI}$ Mean | $A_{0LI}$ Std | $A_{0KR}$ Median | $A_{0KR}$ Mean | $A_{0KR}$ Std |
|---|---|---|---|---|---|---|---|---|---|---|---|---|---|
| 1 | 0.40 | 1097 | 1076 | 299 | 229 | 260 | 131 | 232 | 288 | 195 | 376 | 385 | 54 |
| 2 | 0.80 | 1312 | 1424 | 413 | 171 | 176 | 93 | 150 | 188 | 128 | 547 | 559 | 150 |
| 3 | 0.85 | 691 | 744 | 288 | 202 | 220 | 123 | 197 | 249 | 169 | 318 | 328 | 35 |
| 4 | 0.80 | 1200 | 1259 | 242 | 258 | 290 | 101 | 239 | 295 | 167 | 489 | 479 | 61 |

**Table 2.** Range of seismic properties assumed for the lower ice shelf. $\nu$ denotes Poisson's ratio.

| | $V_p$ (m s$^{-1}$) | $V_s$ (m s$^{-1}$) | $\rho$ (kg m$^{-3}$) | $\nu$ |
|---|---|---|---|---|
| Debris laden ice | 3800–3870 | 1930–2040 | 917–1274 | 0.297–0.330 |
| Water | 1450 | 0 | 1028 | |

the known reflector method and the resulting $A_0$ estimates are shown in Figure 4 (right column) and Table 1 ($A_{0KR}$ columns). The method resulted in the same number of $A_0$ estimates as the multiple bounce method and each line's average basal ice properties estimated during optimisation are shown in Table 3. The known reflector method requires an estimate of path effects but is insensitive to our assumption that $\alpha = 0.27$ km$^{-1}$ as the same $\alpha$ used to determine $A_0$ is later used in Equation 6 to
determine the basal reflection coefficient. The known reflector method has similarities to the technique used by Smith et al. (2018) in their study of the lithology beneath Subglacial Lake Ellsworth, although here we explicitly estimated $A_0$, allowed the basal ice properties to vary, and used amplitude versus offset techniques.

## 2.5 Choosing the best $A_0$

The known reflector method provided our best estimate of $A_0$ as judged by its potential to recover accurate estimates of basal
reflectivity (e.g. ice–water reflection coefficient where the ice is known to be floating), and its narrow normal distribution (Figure 4, Table 1). The narrow distribution indicates low source size variability, consistent with a uniform firn–ice profile, a consistent drilling depth and geophone placement, back filling all shots, and allowing at least 24 hours before detonation.

Both our direct path methods resulted in large standard deviations (Table 1) and correlate poorly with our known reflector estimates ($r^2$ (coefficient of determination) of 0.09 for the direct pair method and 0.04 for the linear intercept method, Figure 5).
The linear intercept method resulted in an average $\alpha = 1.4 \pm 0.5$ km$^{-1}$ (mean and 1 standard deviation of the combined results for all 4 lines). Individual line average values range from 1.0–1.6 km$^{-1}$. These $\alpha$ estimates are an order of magnitude greater than commonly used published estimates and are not used in our analysis. The multiple bounce method correlates well with

**Table 3.** Seismic properties estimated in the lower ice shelf

|        | $V_p$ (m s$^{-1}$) | $V_s$ (m s$^{-1}$) | $\rho$ (kg m$^{-3}$) | $\nu$ | %Debris |
|--------|------|------|------|------|------|
| Line 1 | 3830 | 1990 | 1030 | 0.31 | 6 |
| Line 2 | 3840 | 1990 | 1030 | 0.32 | 7 |
| Line 3 | 3830 | 1990 | 1030 | 0.31 | 6 |
| Line 4 | 3850 | 1960 | 1030 | 0.33 | 6 |

the known reflector method ($r^2$=0.46, Figure 5). Linear regression of the known reflector estimates with the multiple bounce estimates results in a best fitting gradient of 2.2 with an intercept of 180. However, this relationship is dependent on our estimate of $\alpha$ and our $\gamma$ estimates, and will be discussed in Section 4.

## 2.6 Estimating subglacial properties

Using each line's $A_0$ values from the known reflector method (Table 1, Figure 4 right column) we calculated the angle dependent bed reflection coefficients for each shot gather ($R(\theta)$, Equation 1). Our angle coverage typically extends up to 25°, with some shots extending to 30°. We present $R(\theta)$ as average values within 10 degrees of normal incidence ($R_b$) (Figures 6A, 7A) to allow comparison with normal incidence methods reported elsewhere (e.g. Muto et al., 2019). We then calculated the optimal combination of subglacial seismic velocities ($V_p, V_s$) and density ($\rho$) (Figures 6– 7B–D) by fitting each shot's entire $R(\theta)$ to the Zoeppritz equations while imposing reasonable bounds for the subglacial material following Zechmann et al. (2018), expanded to allow for an ice/water interface (Table 4). During optimisation we imposed the additional constraint that the optimal $V_p$ and $V_s$ must result in a realistic Poisson's ratio ($\nu$) of 0.25–0.5 (Hamilton, 1979). Optimisation minimised the root mean squared misfit between the observed amplitudes for each shot and those modelled by the Zoeppritz equations using the *fmincon* algorithm in MATLAB®. This optimisation uses a trust region approach resulting in rapid convergence. We set the basal ice's seismic properties to those obtained for each line during our $A_0$ known reflector method in Section 2.4.3 (Table 3). We repeat our $R_b$ estimates and the optimisation of $V_p$, $V_s$, and $\rho$ values using $R(\theta)$ values estimated using all estimates of $A_0$ for each line, resulting in the same number of estimates of basal properties per shot as there are estimates of known reflector source size per line. In some cases our inversion repeatedly converged on the same solution implying a misleadingly high precision. To account for this we also estimated our uncertainties by examining the retrieved basal properties from the floating portions of our survey. For all floating portions of the survey, misfit between the recovered properties and theoretical properties resulted in one standard deviation uncertainties for $R_b$ of ± 0.09, $V_p$ of ± 140 m s$^{-1}$, $V_s$ of ± 430 m s$^{-1}$, and $\rho$ of ± 30 kg m$^{-3}$. Uncertainty estimates for each line are shown in Table 5.

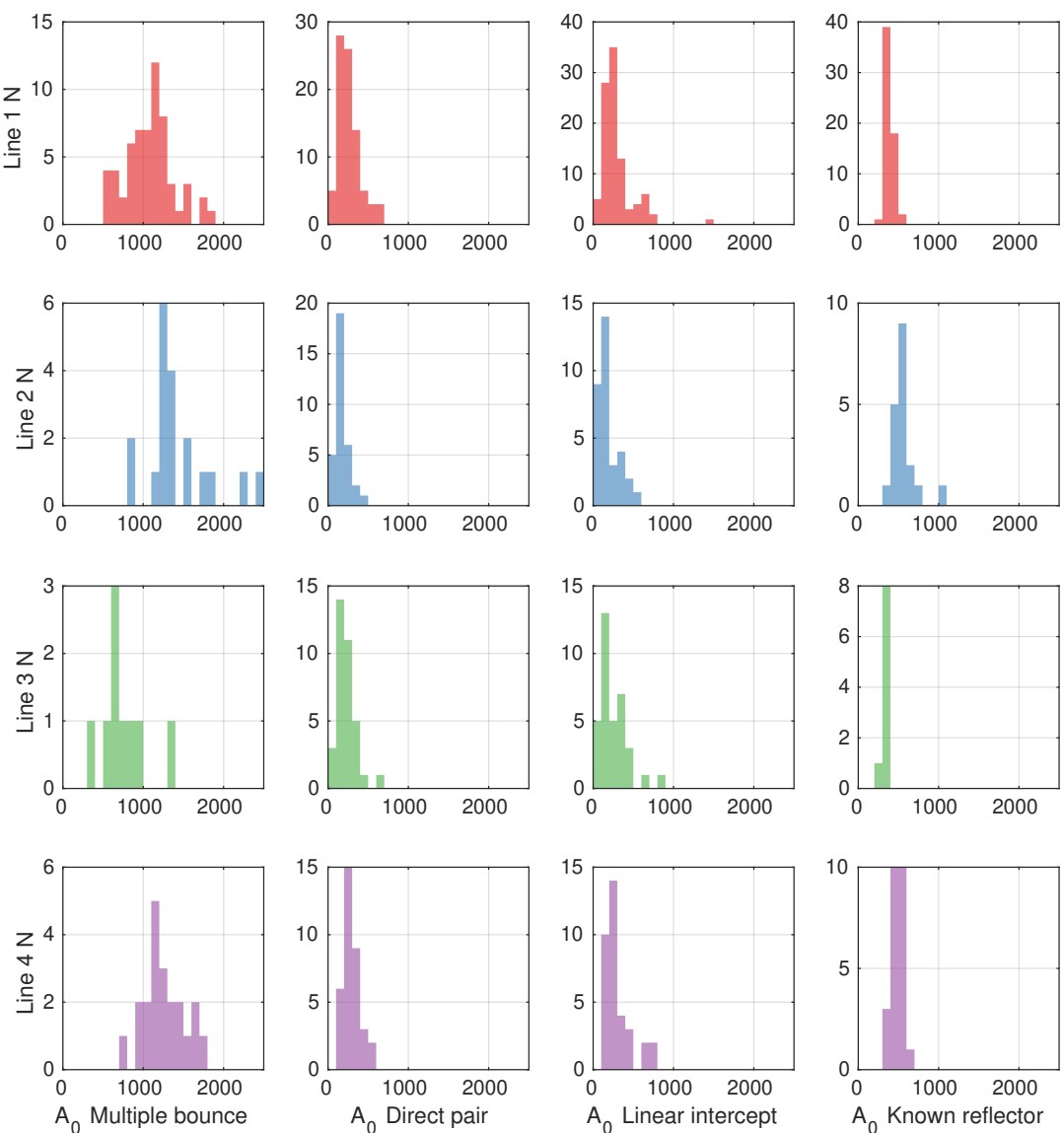

**Figure 4.** $A_0$ source size estimates for Whillans Grounding Zone Lines 1–4 (rows) using four methods (columns). Left column: $A_0$ estimates using the primary–multiple amplitude ratio method. Centre left column: $A_0$ direct pair estimates. Centre right column: $A_0$ linear intercept estimates. Right column: $A_0$ estimates from known reflection coefficient method assuming ice overlying water. (See Figure 1 for line locations.)

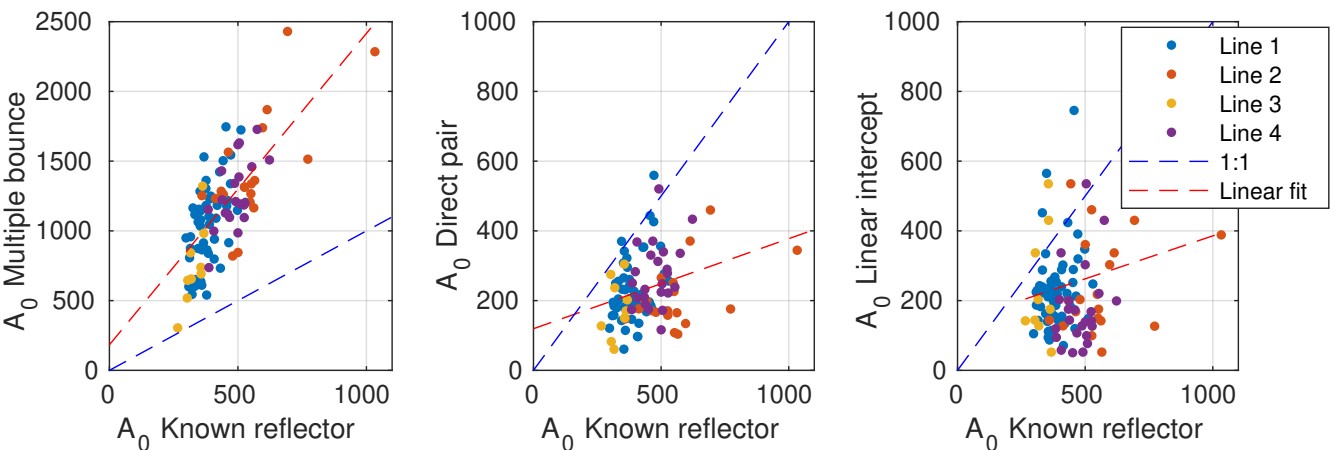

**Figure 5.** $A_0$ estimates comparisons. Left: $A_0$ estimates from known reflector method against $A_0$ estimates from multiple bounce method (coefficient of determination ($r^2$) of linear regression = 0.45). Middle: $A_0$ estimates from known relectivity method against $A_0$ estimates from the direct pair method ($r^2$=0.09). Right: $A_0$ estimates from known reflector method against $A_0$ estimates from linear intercept method ($r^2$=0.04).

**Table 4.** Seismic velocity ($V_p$, $V_s$), density ($\rho$) and Poisson's Ratio ($\nu$) bounds used for Zoeppritz fitting.

| | |
|---|---|
| $V_p$ (m s$^{-1}$) | 1440–2300 |
| $V_s$ (m s$^{-1}$) | 0–1150 |
| $\rho$ (kg m$^{-3}$) | 1000–2500 |
| $\nu$ | 0.25–0.5 |

## 3 Results

### 3.1 Reflection Coefficients and Basal Properties

Line 1 exhibits generally slowly varying $R_b$ values upstream of the grounding zone, before an abrupt change at the grounding zone (Figure 6). This change occurs over less than 500 m at approximately kilometer 9. $V_p$, $V_s$ and $\rho$ values retrieved from
5   Zoeppritz fitting exhibit a similarly abrupt change at the grounding zone. Upstream of the grounding zone binned mode $V_p$ and $V_s$ values equal 2000 m s$^{-1}$ and 1100 m s$^{-1}$ respectively and mode $\rho$ values equal 1800 kg m$^{-3}$. Kilometer 3–4 of Line 1 exhibits retrieved $V_s$ and $\rho$ values similar to those expected for water, but $R_b$ and $V_p$ estimates suggest otherwise. In the floating portion of the profile most retrieved properties are equal those expected for water (Table 5). Estimates of $V_s$ are more spatially variable with larger distributions both upstream and downstream of the grounding zone.
10   Line 2 exhibits similar patterns in $R_b$ and retrieved seismic properties to Line 1. An abrupt transition is observed at the grounding zone (kilometer 3.6), and the grounded and floating portions are dominated by distinct seismic properties (Figure 7,

**Table 5.** Binned mode seismic properties estimated using normal incidence methods ($R_b$) and Zoeppritz fitting ($V_p$, $V_s$, and $\rho$) for the grounded and floating portion of each line. Bin sizes are shown in square brackets. One standard deviation uncertainties were obtained from the misfit in the floating portion of each line.

|  | $R_b$ [0.05] | $V_p$ (m s$^{-1}$) [50] | $V_s$ (m s$^{-1}$) [100] | $\rho$ (kg m$^3$) [25] |
|---|---|---|---|---|
| Line 1 Grounded | -0.10±0.09 | 2000±140 | 1100±300 | 1800±30 |
| Line 1 Floating | -0.45±0.09 | 1450±140 | 0 (0–300) | 1000±30 |
| Line 2 Grounded | -0.10±0.14 | 2000±150 | 1100±830 | 1675±30 |
| Line 2 Floating | -0.40±0.14 | 1450±150 | 0 (0–830) | 1000±30 |
| Line 3 Grounded | -0.20±0.08 | 2000±70 | 1100±330 | 1000±30 |
| Line 3 Floating | -0.45±0.08 | 1450±70 | 0 (0–330) | 1000±30 |
| Line 4 Grounded | -0.10±0.09 | 2000±130 | 1100±630 | 2000±30 |
| Line 4 Floating | -0.45±0.09 | 1450±130 | 0 (0–630) | 1000±30 |

left panel), Table 5). Upstream of the grounding zone two retrieved estimates exhibit properties similar to those of water (kilometer 0–0.5); however, neither are unambiguous. $V_s$ estimates are again more variable than other parameters, with most floating shots exhibiting $V_s$ values typical of water. Line 3 (Figure 7, middle panel) shows both rapid and gradual changes in basal properties along the profile. Rapid changes are observed either side of kilometer 7–8 where a narrow bed feature exhibits

$V_p$ and $\rho$ estimates typical of subglacial water. Kilometer 2–4 displays a gradual change in $R_b$ while the associated transition in $V_p$ and $\rho$ occurs abruptly over <500 m. $V_s$ estimates are variable along the profile, and exhibit scatter within regions thought to be both grounded (kilometer 0–3) and floating (kilometer 3.5–6). Line 4 (Figure 7, right panel) is dominated by $R_b$, $V_p$ and $\rho$ estimates typical of ice over water (kilometer 0–7). The transition from these values occurs over a distance of <1 km beginning at kilometer 7. As with the other profiles the estimates of $V_s$ are variable but most often the floating portion of the

profile (kilometer 0–7) exhibits $V_s$ estimates typical of water (Table 5).

## 3.2    Experiment Timing and Tidal Elevation

Seismic shooting occurred at different stages of the tide resulting in the potential for different tidal heights along profile. Shot and receiver elevations were not directly observed at the time of shooting so instead we present tidal heights estimated at the floating end of the profile using Erofeeva et al. (2020) (Figure 8). Figure 8A shows that kilometer 6–12.5 of Line 1 was acquired

on the falling tide when the tidal elevation varied from +0.1 m to -0.6 m. The pronounced change in basal reflectivity that occurs at approximately kilometer 9 on Line 1 (Figure 6) does not coincide with a step in the tidal elevation. Other step-changes in tidal elevation along Line 1 also do not coincide with changes in basal reflectivity (e.g. kilometer 1, 6). Lines 2–4 all took less than a day to acquire and for the most part have no major step-changes in tidal elevation along the profiles. An exception to

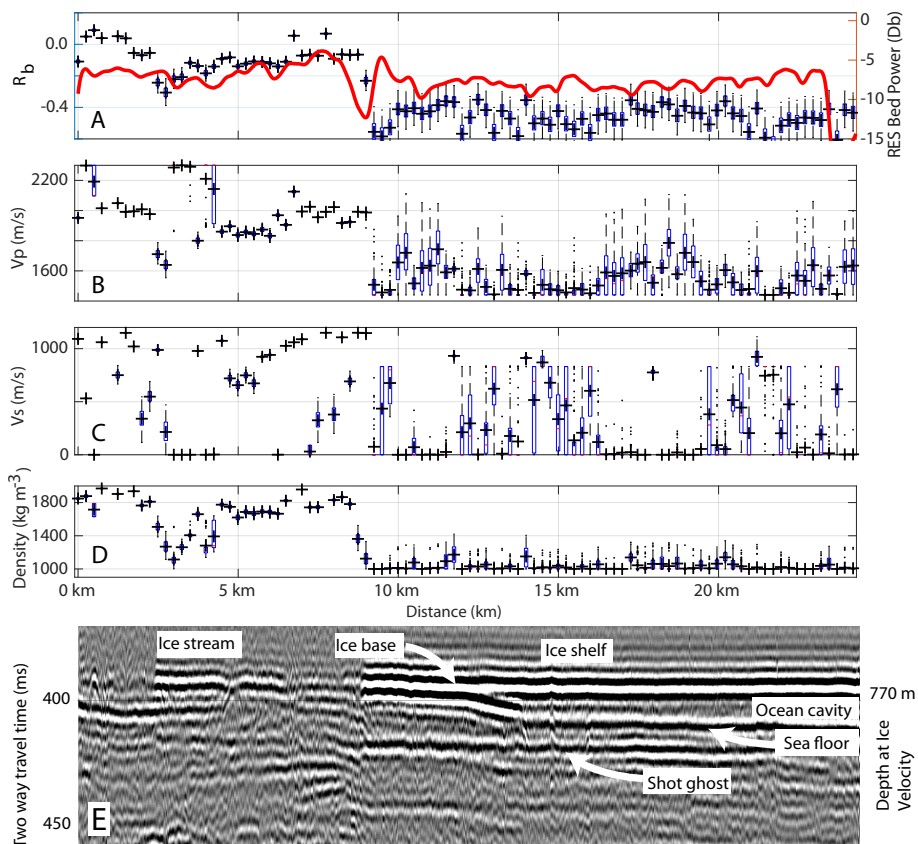

**Figure 6.** Line 1 (A) Seismic basal refectivity at normal incidence estimated from the average value within $10°$ ($R_b$). Red line shows radar basal reflectivity from Christianson et al. (2016). (B–D) Box plots of $V_p$, $V_s$ and $\rho$ estimated using Zoeppritz fitting and all estimated source sizes. Blue boxes show the 25th and 75th percentiles, whiskers extend to cover data points, and outliers are plotted as black points. Solutions using the mean source size are overlain as black crosses. All estimates use source sizes obtained using the known reflector method. (E) Stacked active source seismic reflection profile with ice flow from left (grounded ice stream) to right (floating ice shelf). Shot ghost denotes the short-path multiple generated by the ray path from the source up to the ice-air interface then down. For profile location see Figure 1.

this occurs on Line 2 where the onset of high basal reflectivity (kilometer 3.6–4.1, Figure 7 left panel) occurs in proximity to an offshore 0.3 m change in tidal elevation.

### 3.3 Repeat elevation profiles across the grounding zone.

Repeat kinematic GNSS elevation profiles were acquired along Lines 1 and 2 (Figure 9) and have previously been used to validate the seismically imaged grounding line location (Horgan et al., 2013b). We locate the grounding zone using the standard deviation of elevation observations in 50 m spatial bins after the removal of a single best fitting spline from each profile. Upstream of the grounding zone we expect this value to represent the method uncertainty, which comes from both

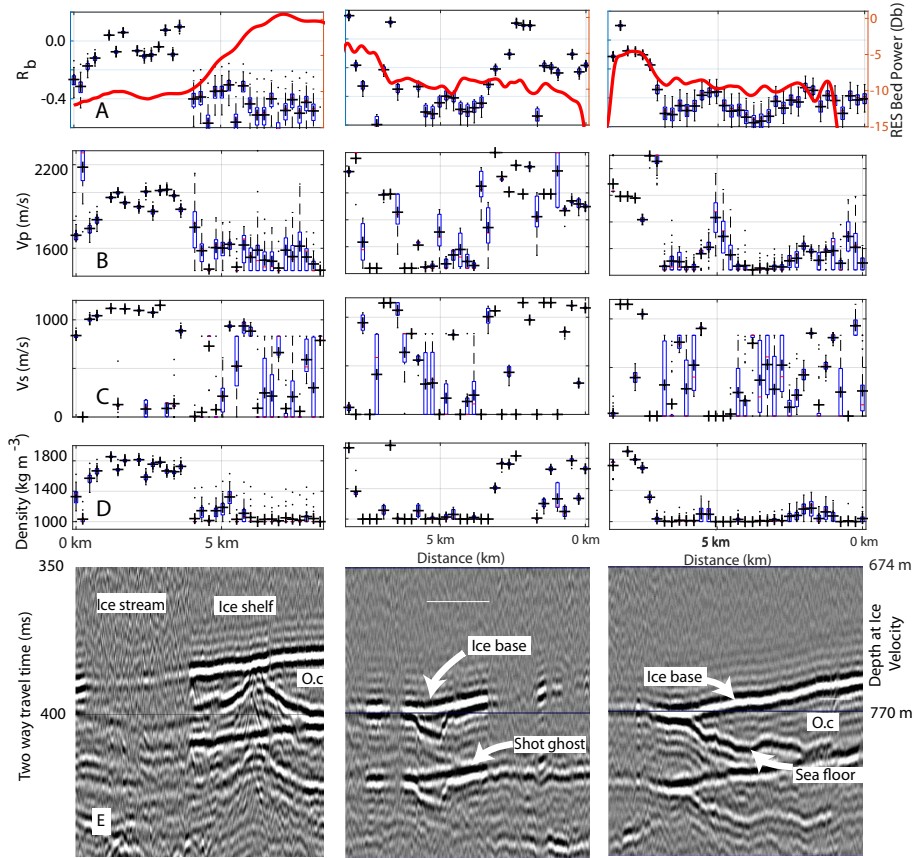

**Figure 7.** Lines 2 (left), 3 (middle), and 4 (right).(A) Seismic basal refectivity at normal incidence estimated from the average value within $10°$ ($R_b$). Red line shows radar basal reflectivity from Christianson et al. (2016). (B–D) Box plots of $V_p$, $V_s$ and $\rho$ estimated using Zoeppritz fitting and all estimated source sizes. Blue boxes show the 25th and 75th percentiles, whiskers extend to cover data points, and outliers are plotted as black points. Solutions using the mean source size are overlain as black crosses. All estimates use source sizes obtained using the known reflector method. (E) Stacked active source seismic reflection profile. Line 2 is plotted flowing from grounded (left) to floating (right). Lines 3 and 4 are plotted with flow into the page. Shot ghost denotes the short-path multiple generated by the ray path from the source to the ice-air interface then down. O.c. denotes the ocean cavity. For locations see Figure 1.

the GNSS observations and our ability to repeat a track precisely, combined with a measure of the roughness of the surface. Downstream these combine with the displacement of the ice surface due to the tide. The grounding line is determined to be the point at which the standard deviation changes from values representative of grounded upstream values to those representative of floating values. The pick is subject to some interpretation as roughness and the ability to repeat a track can vary spatially

5    and can correlate with surface slope (e.g. van der Veen et al., 2009).

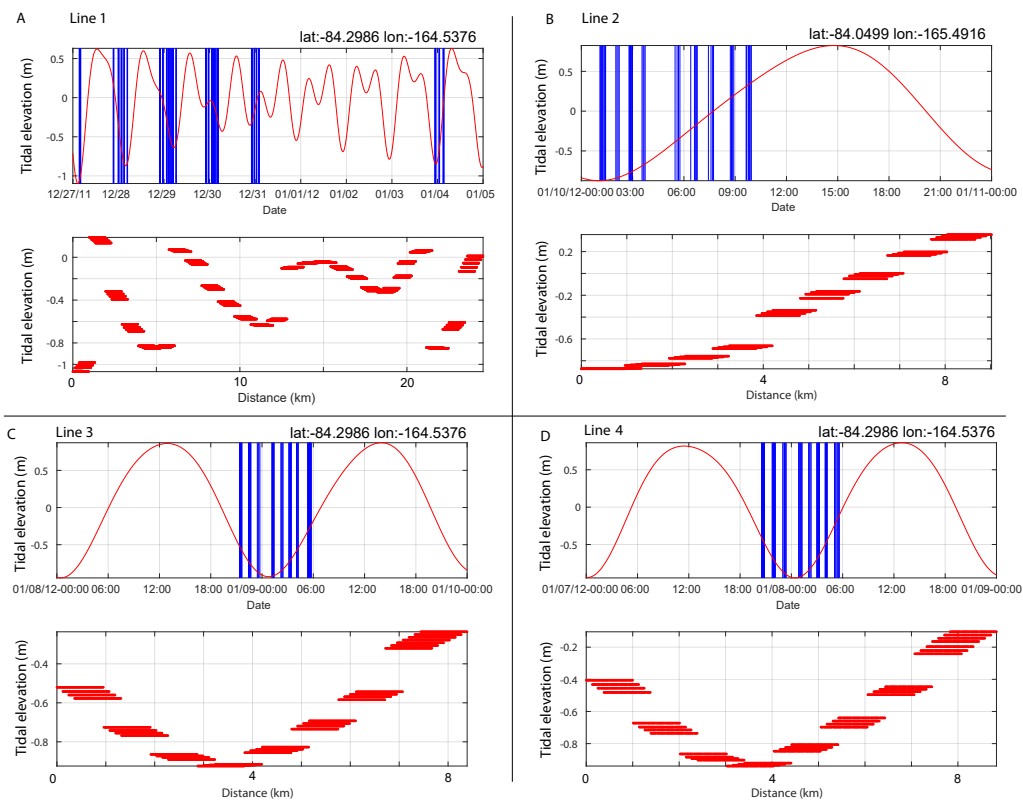

**Figure 8.** Shot timing and tidal elevation from Erofeeva et al. (2020). A) Line 1. Top subplot shows the timing of shots (blue bars) overlain on the tidal elevation anomaly. Bottom subplot shows vertical tidal anomalies (Erofeeva et al., 2020) at the time of shooting as a function of distance along the profile. B-C) same as A) but for lines 2, 3, and 4, respectively. Latitude (lat) and longitude (lon) for each tide model time series is shown in each top subplot.

Repeat elevation profiles for lines L1 and L2 were acquired on the rising tide. The tidal range for Line 1 at the time we observed was approximately 1.5 m, while Line 2 was observed during a range of approximately 0.35 m. Both profiles exhibit a region of relatively-high surface slope that begins upstream of the onset of vertical tidal displacement. We pick the Line 1 grounding line at kilometer 9.6, and at kilometer 3.6 for Line 2. Well upstream of the grounding zone our repeat tracks typically
5    all fall within 0.1 m vertically of each other. At the resolution of our data we do not observe migration of the grounding line in the GNSS data.

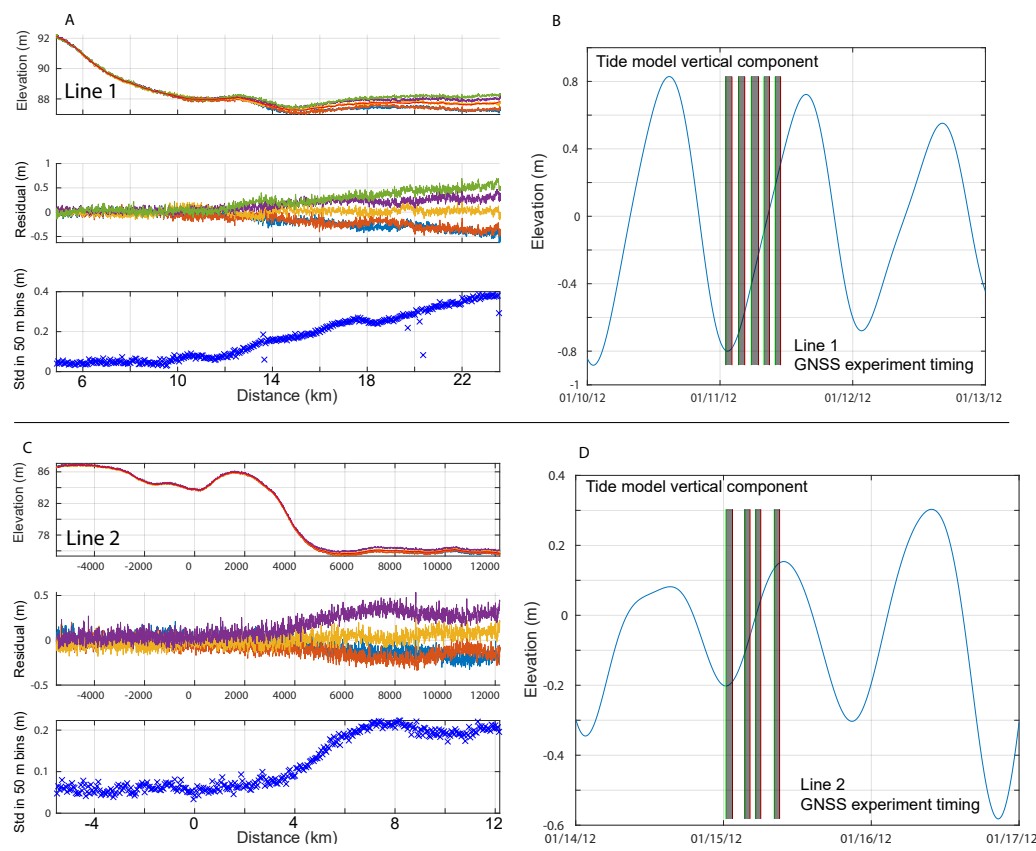

**Figure 9.** Repeat kinematic profiling along Lines 1 (A,B) and 2 (C,D). Left panels (A,C) show the elevation (top), residual elevation after removal of a best-fitting spline (middle), and standard deviation of residual elevation in 50 m spatial bins (bottom). Panels B,D show the timing of the GNSS profile data collection (vertical overlain on the vertical elevation anomaly of Erofeeva et al. (2020).

## 4 Discussion

### 4.1 Subglacial properties beneath Whillans Ice Stream's grounding zone

Subglacial material beneath the grounded ice stream exhibits $\rho$ and $V_p$ values in the range of dilatant till, but with most $V_s$ values typical of those observed in dewatered tills (Figure 6–7, Table 5) (Zechmann et al., 2018). Our estimates of $V_p$ and $\rho$ for
5  all lines are close to those estimated by Luthra et al. (2016) in their active source seismic study of a major sticky spot beneath Whillans Ice Plain. $V_s$ estimates from the grounding zone are greater than those estimated by Luthra et al. (2016), although they overlap within uncertainties. When compared with estimates from upstream on Whillans, where Blankenship et al. (1986) measured $V_s$ of $150\pm10$ m s$^{-1}$, our results indicate significantly stiffer till beneath the grounding zone. Basal shear stress is already known to vary spatially beneath Whillans Ice Stream. Inversion of surface elevation, ice thickness, and remotely-sensed
10  velocity observations has resolved spatially variable basal shear stress (Joughin et al., 2004b), and spatially variable rates of

change of basal shear stress during the ice stream's deceleration (Beem et al., 2014). Joughin et al. (2004a) estimated low basal shear stress near the grounding zone, similar to that observed elsewhere beneath the majority of the ice plain. Lipovsky and Dunham (2017) introduced spatial variable bed properties in their rate and state friction model to better reproduce the timing and distribution of stick–slip displacement on Whillans Ice Plain. Passive seismic and geodetic observations of Whillans Ice

Stream's stick-slip motion have been used to locate asperities beneath the central portion of the ice stream (Walter et al., 2011) and at its grounding zone (Pratt et al., 2014).

The transition in basal properties at the grounding zone of Whillans Ice Stream is abrupt in both longitudinal lines (Lines 1–2), occurring over distances of less than 500 m. This is less than the ice thickness of 730–790 m. The transverse lines (Lines 3–4) exhibit less abrupt transitions but still show change over distances of less than 1 km. The rapid transition in basal properties

suggest that even in the case of a fast flowing, low basal shear stress ice stream such as Whillans, it is necessary to solve the full Stokes equations if the ice flow velocity field is to be accurately modelled across the grounding line (Pattyn et al., 2013). The radio echo sounding (RES) results of Christianson et al. (2016) provide additional insights (Figures 6A, 7A). Lines 1,3, and 4, which all sample the embayment in the grounding zone to the grid north (Figure 1), all exhibit a drop in RES basal reflectivity of approximately 3–5 dB as the grounding zone is crossed from the grounded ice stream to the floating ice shelf. This change

occurs over similar length scales to the seismically detected transition. In contrast, Line 2, which crosses the peninsula to the grid south exhibits a gradual increase in RES basal reflectivity of approximately 10 dB after the ice goes afloat, over a distance of approximately 3 km. Christianson et al. (2016) attributes the differences in the RES-detected transitions to the presence of basal roughness (fluting, modelled with a 20 m wavelength and 4 m root-mean-squared heights) and entrained debris in the ice shelf in the embayment, and a basal interface that is becoming smoother and losing the basal debris zone due to basal melt

at the peninsula. The percentage of entrained debris we obtained during source size estimation is similar across all four lines (6–7%), indicating differing debris content is unlikely to be the cause of the differences in RES basal reflectivity. MacGregor et al. (2011) reported low frequency (2 MHz) RES bed reflectivity from elsewhere on Whillans Ice Stream and the adjacent Kamb Ice Stream and found negligible change in RES reflectivity when crossing the grounding zone. One possibility discussed by MacGregor et al. (2011) was the presence of brackish water upstream of the grounding line, smoothing the RES-imaged

transition from grounded to floating ice.

## 4.2   Water upstream of the grounding zone

Upstream of the grounding zone several regions (e.g. Line 1 kilometer 3–4; Line 2, kilometer 0–0.5; Line 3 kilometer 7–8) exhibit properties that indicate the presence of subglacial water, although not without ambiguity. This ambiguity likely results from water column thicknesses that are less than one-quarter the dominant seismic wavelength for our data ($\lambda/4 \approx$

3.6 m). Visual inspection of shot records shows that in these regions the thin-layer effects detailed by Booth et al. (2012) result in constructive and destructive interference of our basal wavelet, leading to best-fitting parameter combinations that are not representative of the contrast in properties. A similar phenomenon likely results in the anomalous estimated values at the grounding zone of Line 1 (Figure 6, kilometer 9) and for kilometer 7–7.5 of Line 3 (Figure 7). However, no similar attribution

is possible for the $V_s$ outliers in the floating portions of all lines, which instead appear to correspond to low signal to noise ratios apparent in visual inspection of the shot records.

Our seismic methods are insensitive to whether the subglacial water is sourced from beneath the ice stream or from the ocean cavity. The WISSARD field site was initially selected as it lay on the subglacial drainage path from Subglacial Lake Whillans to the ocean cavity (Fricker et al., 2010; Carter and Fricker, 2012). Oversnow geophysical surveying, including the data presented here and in Christianson et al. (2013), has shown the potential for estuarine flow across the grounding zone (Horgan et al., 2013a). Shot times, tidal stage, and bed reflectivity lack correlation between changes in tidal height and imaged bed properties. One exception to this occurs on Line 2 where the change in bed properties at kilometer 3.6 (Figure 6 left column) occurs in proximity to a 0.3 m change in tidal height at kilometer 3.8–4.2 (Figure 8B). We consider this correlation coincidental as Line 2's grounding line position appears pinned at kilometer 3.6 by an approximately 6 m change in bed elevation. Also, repeat GNSS profiling (Figure 9C) indicates vertical change at Line 2's grounding line is likely to be much less than that estimated offshore, and even a 0.3 m change in water column thickness would be insufficient to cause the pronounced change in reflectivity observed. Line 1's repeat GNSS profiling (Figure 9A) locates the onset of vertical tidal deflection 0.6 km downstream of the seismically resolved change in subglacial properties. This indicates the presence of water upstream of the of the GNSS picked grounding line, but the subjective nature of the GNSS method make this conclusion tentative. Line 1's repeat GNSS profiling also suggests the region between kilometer 9.6–12 is a zone of ephemeral grounding, resulting in a smaller distribution of elevations over the observed portion of the tidal cycle (Figure 9A bottom subplot). Our experiment was not designed to study changing bed properties over a tidal cycle, which would be better examined using tilt meters or fixed GNSS stations and a fixed geophone deployment with a source repeating at the same location.

While our methods are not able to determine the process of stiffening at the grounding zone and ponding upstream, our observations are broadly consistent with the findings of several previous modelling studies. In the nomenclature of Sayag and Worster (2013) our study location appears to be a fixed grounding line, stiff-bedded system, although the zone of emphemeral grounding and the 0.6 km difference between our seismically determined grounding zone and that located by our repeat GNSS profiling shows some grounding line migration may be occurring on Line 1. Our seismic properties indicate a stiff bed over thicknesses of at least approximately 5 m ($\lambda/4 = 5$m for a 100 Hz wave in a 2000 m s$^{-1}$ medium). Estimated seismic velocities and densities imply a Young's modulus ($E$) of 3.1–6.2 GPa in the subglacial material with lines 1,2, and 4 all exhibiting $E =$5.2–6.2 GPa. Our observations at this location are not able to identify the asymmetric grounding line migration outlined by Tsai and Gudmundsson (2015). Local variations in bed and surface slope, and ice thickness are likely to contribute to this, however the resolution of our GNSS method and our temporal sampling of basal properties also contribute to a lack of fidelity. Stiff till beneath the grounding zone and localised bodies of water upstream of the grounding zone are in keeping with the compression and dewatering of subglacial till due to ice flexure modelled by Walker et al. (2013). Stiffening of the till was also invoked by Christianson et al. (2013) as the cause of the enhanced internal deformation evident in radio echo sounding profiling across the grounding zone. The presence of isolated water bodies also aligns with the alternating pressure gradients causing barriers to water flow upstream of the grounding proposed by Sayag and Worster (2013) and the movement of water upstream of the grounding line modelled by Warburton et al. (2020). Warburton et al. (2020) show that low subglacial

permeability should lead to filtering of the response of ice flow to tidal forcing. If this is true for Whillans Ice Stream then the combination of the low till permeability suggested by our findings, and the tidally modulated twice daily stick–slip motion of the ice stream indicates its response to tides is not controlled by fluid connectivity through the grounding zone till.

### 4.3   Estimating source size ($A_0$)

Our preferred method of estimating source size is only possible when a portion of the survey area contains a known reflection interface. The interface need not be known exactly, as demonstrated by our retrieval of basal ice properties alongside estimating source size, provided the shape of the $R(\theta)$ response varies with changing properties along with the absolute level of reflectivity. Comparison with other methods used to estimate $A_0$ demonstrates the efficacy of the commonly employed multiple bounce method (Figure 5). $A_0$ estimated using the multiple bounce method was, however, approximately twice that estimated using

our known reflector method (Figure 5). This difference can be reduced by a more thorough treatment of the path amplitude factor ($\gamma_i$). For instance, applying the geometric loss estimated by Margrave (2003) results in a best fitting gradient of 1.6. The remaining difference can be accounted for by varying $\alpha$ in our known reflectivity $A_0$ calculation, with an $\alpha = 6.0$ km$^{-1}$ resulting in a 1:1 relationship between the multiple bounce and known reflector methods, albeit with a linear intercept of approximately 100. Instead of using path amplitude factors from Margrave (2003) and adjusting our $\alpha$ estimate we have

chosen the 1/pathlength approach of Equation 2 and a published $\alpha$ estimate for clarity and to better enable repeatability. The discrepancy between the methods indicates that attenuation ($\alpha$) and path amplitude factors ($\gamma$) remain areas of uncertainty, overcome here by our use of a known reflector. In the absence of reliable $A_0$ estimates, other attributes of the amplitude reflection curve such as the angle of phase change (e.g. Anandakrishnan, 2003a) can be effective predictors of subglacial geology. Direct path methods for $A_0$ estimation have been successfully employed elsewhere (Muto et al., 2019), and greatly

simplify $R(\theta)$ recovery. Muto et al. (2019) presented data where the sources were buried at 40–50 m depth, compared to our 27 m, and their signal to noise ratios are high as evident in their imaging of englacial seismic reflectivity. The poor correlation between our known-reflector and direct-path $A_0$ estimates (Figure 4) shows that further investigation of direct path methods is warranted. Both the direct path methods we present would benefit from a greater offset distribution, and the direct pair method would benefit from a greater number of path combinations where $s_2/s_1 = 2$ than was available to us. Trace interpolation could

also be used here as the direct arrival energy is unlikely to change rapidly. Also, the path effects ($\gamma_{d,i}$) experienced by the direct ray are likely to be inadequately captured by our approach due to the possibility of unaccounted for energy loss and more complex travel paths than those predicted within the firn.

Our Zoeppritz fitting methodology is skilled at recovering both $V_p$ and $\rho$ as demonstrated in the floating portions of all lines where the recovered values are those expected for water (see Table 5 floating estimates). The methodology is less skillful at

recovering $V_s$, likely due to the weaker dependence of the shape of the $R(\theta)$ curve on $V_s$ for the angles we observe. Using average source sizes and the known reflector method we recover the near zero $V_s$ typical of water for 73 of the 112 floating shots in our survey. Estimating $V_s$, along with $\rho$ allows the shear modulus to be estimated, which can be used to calculate the effective pressure in the till (Luthra et al., 2016). This provides a more direct link between seismic observations and till properties than is otherwise possible from estimates of normal incidence reflectivity ($R_b$) alone. An acquisition geometry that covered greater

angles would improve our ability to estimate $V_s$; however, limitations due to interference from direct arrivals would still exist. These limitations could be overcome by observing much greater offsets, where direct arrivals no longer interfere with the bed return, or surveying in regions of greater ice thickness. Using multiple charge sizes and configurations also highlights the importance of source configuration. Line 3, which consisted of the largest charges by weight (0.85 kg) resulted in the lowest

$A_0$ estimates calculated from both the known reflector method and the multiple bounce method. The charges for Line 3 were made up of a stack of a single 0.4 kg charge, and three narrower 0.15 kg charges. These narrower charges were likely less well coupled with the shot hole wall, and the longer linear configuration resulted in a less effective source. A shorter interval between shot loading and detonation may have also been a factor here as Line 3 was shot within 1–2 days of loading.

## 5   Conclusions

Subglacial material beneath Whillans Ice Stream's grounding zone is relatively stiff exhibiting $V_s \approx 1100$ m s$^{-1}$ and Young's moduli of 3.2–6.2 GPa, making it more similar to a subglacial sticky spot than to deforming till. The transition from this stiff subglacial sediment to the ocean cavity is abrupt, occurring over distances of 500-1000 m. This seismically imaged transition differs from that imaged using RES, which detects both an abrupt transition and a gradual one at the embayment and promontory respectively (Christianson et al., 2013). Upstream of the grounding line we detect thin, apparently isolated, bodies

of water. These findings are consistent with models that compact till within the grounding zone and those that isolate water upstream of the grounding zone, although we cannot detect whether the subglacial water is sourced from the ocean cavity or subglacially. Our comparison of methods used to determine source size ($A_0$) shows that the commonly employed multiple bounce method correlates well with the known reflector method available to us. However, our comparison also highlights that path effects ($\gamma_i$) are incompletely modelled by the methods employed here and elsewhere. Our findings also reinforce the need

for consistency in source placement, configuration, and time between burial and detonation. Overall our methods are skilled at retrieving basal properties at relatively high spatial resolution where the thickness of the subglacial material is sufficient to prevent thin film effects ($> \lambda/4$). Both $V_p$ and $\rho$ are reliably retrieved, while $V_s$ is recovered less consistently. While we are currently unable to accurately recover all seismic properties for what appear to be thin water layers, our methods do show promise here. These thin layers are pertinent for ice flow, and techniques such as full waveform inversion are likely to prove

useful here. These methods, which invert not just for a single amplitude of the basal return but the full time series, have been successful applied to other environments where thin layers with large contrasts in seismic properties have been investigated (e.g. Pecher et al., 1996).

*Code and data availability.* Data analysis and modelling used MATLAB® and the CREWES Matlab Toolbox (www.crewes.org). Seismic data processing and picking was performed using GLOBE Claritas (www.globeclaritas.com). Amplitude data are available at PANGEA

(www.pangaea.de/ DOI pending.)

*Competing interests.* No competing interests are present.

*Acknowledgements.* We are grateful to Rory Hart, Matthew Hill, and Benjamin Petersen for assistance in the field. HJH thanks Roger Clark for helpful correspondence early in this study. This study was funded by US National Science Foundation grants to the CReSIS (0852697), and WISSARD (0838764, 0838763) projects. HJH acknowledges funding from a Rutherford Discovery Fellowship and Project-1 of the New
5 Zealand Antarctic Science Platform. An anonymous reviewer, Alex Brisbourne, and editor Reinhard Drews are thanked for their comments.

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
