# Peer review of "Grounding zone subglacial properties from calibrated active source seismic methods"

_The Cryosphere, 2020_

## Referee Comment (RC1) · Anonymous Referee #1 · 27 Jul 2020

The authors present an active seismic study in the grounding zone of the Whillans Ice Stream. Although their seismic methods are interesting, the paper needs a lot of work. Basic hypotheses (i.e., the influence of tides) about their findings are unexplored. Their clustering analysis is threadbare. Most lacking is the intellectual context. Other groups (besides folks from Penn State) do work on Whillans and it would benefit progress in the field if the authors showed more interest in interacting with these other lines of inquiry. Despite these criticisms, I do think that a significantly modified manuscript could meet publication standards in the Cryosphere.

**Data, Methods, Results**

First major point: the clustering warrants more detail. How was the number of clusters chosen? It would be useful to see an L-curve analysis to persuade the reader that

three is a reasonable number. Were error bounds included in the cluster analysis? It would be interesting to do so. It would also be interesting to see the clustering carried out with Vs included for the following reason. A reasonable null hypothesis would be that there exist two clusters: grounded and floating. These two clusters should be mainly distinguished by Vs. Perhaps the results indicate that grounding zone is more complicated than this simple picture?

2D scatter plots of the clustering results should be shown to demonstrate the validity of the underlying method. See https://scikit-learn.org/stable/modules/clustering.html. Different clustering methods work well with different structures in the data and it's not clear that K-Means is the right choice for this application.

Second major point: Were the shots all fired at the same phase of the tides? See the work by Victor Tsai and others about tidal grounding line migration. Perhaps this is why Line 1 appears to have not hit the ocean cavity? Perhaps the ice was in contact with ocean sediments during the time of acquisition.

Other points:

It would be useful to also report Q values.

Figure 4 (and elsewhere), What are the units of source size?

What is the symbol $R_b$? The text only defines $R_{bInt}$ and $R_{b10}$.

Table 4. Does the plus/minus range indicate one standard deviation?

**Discussion, Conclusions**

I found the discussion to be quite narrow. Whillans is a complex, interesting, and unusual system. The ice stream is decelerating and is expected to be a major stabilizer of Antarctic ice loss. Yet when we go to probe the nature of the deceleration, we see that ice flow is quite bizarre and exhibits a remarkable large scale stick-slip motion. Despite this fascinating situation, the most that the authors can offer in terms of the

implications for ice flow is the already-established point that "The rapid transition in basal properties indicates that the full Stokes equations are likely to be needed to be solved if the ice flow velocity field is to be accurately." The authors also mention the Walker 2013 paper. As written, this paper could be submitted to a seismology-focused journal like BSSA. As indicated in the next section of the review, it would be nicer to see closer integration with other lines of inquiry.

**References**

27 out of 49 of the citations in this paper are to the Penn State group. Of the other 22 citations, the average citation age is in the mid 1990's and none are about the Whillans Ice Stream.

I'm surprised to see no mention of any of the numerous modeling studies that have been carried out on Whillans:

Bougamont, Marion, Slawek Tulaczyk, and Ian Joughin. "Numerical investigations of the slow-down of Whillans Ice Stream, West Antarctica: is it shutting down like Ice Stream C?." Annals of Glaciology 37 (2003): 239-246.

Goldberg, D., C. Schoof, and O. Sergienko (2014), Stick-slip motion of an Antarctic ice stream: The effects of viscoelasticity, J. Geophy. Res. Earth Surf., 119, 1564–1580, doi:10.1002/2014JF003132.

Lipovsky, B. P., and E. M. Dunham (2017), Slow-slip events on the Whillans Ice Plain, Antarctica, described using rate-and-state friction as an ice stream sliding law, J. Geophys. Res. Earth Surf., 122, doi:10.1002/2016JF004183.

Sergienko, O. V., D. R. MacAyeal, and R. A. Bindschadler (2009), Stick–slip behavior of ice streams: Modeling investigations, Ann. Glaciol., 50(52), 87–94.

Similarly for the observational studies of other groups,

Beem, L. H., et al. "Variable deceleration of Whillans Ice Stream, West Antarctica."

Journal of Geophysical Research: Earth Surface 119.2 (2014): 212-224.

Stearns, Leigh A., Kenneth C. Jezek, and Cornelis J. Van Der Veen. "Decadal-scale variations in ice flow along Whillans Ice Stream and its tributaries, West Antarctica." Journal of Glaciology 51.172 (2005): 147-157.

Walter, J. I., E. E. Brodsky, S. Tulaczyk, S. Y. Schwartz, and R. Pettersson (2011), Transient slip events from near-field seismic and geodetic data on a glacier fault, Whillans Ice Plain, West Antarctica, J. Geophys. Res., 116, F01021, doi:10.1029/2010JF001754.

Other shelf/stream papers besides the Penn State Walker et al 2013 paper:

Tsai, Victor C., and G. Hilmar Gudmundsson. "An improved model for tidally modulated grounding-line migration." Journal of Glaciology 61.226 (2015): 216-222.

Sayag, R., and M. Grae Worster. "Elastic dynamics and tidal migration of grounding lines modify subglacial lubrication and melting." Geophysical research letters 40.22 (2013): 5877-5881.

Finally, it is somewhat glaring that there are no citations to the WISSARD project on the Whillans Ice Stream. The authors mention basal "ponding". Other grounds call these ponds subglacial lakes:

Tulaczyk, Slawek, et al. "WISSARD at Subglacial Lake Whillans, West Antarctica: scientific operations and initial observations." Annals of Glaciology 55.65 (2014): 51-58.

Carter, S. P., H. A. Fricker, and M. R. Siegfried. "Evidence of rapid subglacial water piracy under Whillans Ice Stream, West Antarctica." Journal of Glaciology 59.218 (2013): 1147-1162.

Siegfried, Matthew R., et al. "A decade of West Antarctic subglacial lake interactions from combined ICESat and CryoSat‐2 altimetry." Geophysical Research Letters

41.3 (2014): 891-898.

Note that I'm not saying the authors need to cite every one of these papers. Rather, it would simply be nice to see a little bit more interaction with the rest of the intellectual community on topics of interest.

---

## Referee Comment (RC2) · Alex Brisbourne (Referee) · 17 Aug 2020

**Review of "Grounding zone subglacial properties from calibrated active source seismic methods" by H. Horgan et al**

**Alex Brisbourne, August 2020**

The authors present an assessment of active seismic data analysis methods using measurements made at the grounding zone of Whillans Ice Stream. Data cover both grounded and floating ice and therefore present an opportunity to assess and calibrate existing and new data processing methods used to obtain absolute properties of the subsurface. To this end the paper is a useful addition to studies of this nature and builds upon the previous work of Holland and Anandakrishnan (2009) (from here referred to as H&A2009).

The manuscript is well written and structured. However, as outlined in my comments below there are a number of clarifications needed in order that the reader can ascertain exactly how the analysis is applied and how closely this fits with previous work. The methodology description in insufficient in places and clearer self-referencing would improve the readers' ability to follow the methodology.

**General comments**

Section 2.6 - Estimating subglacial properties – Optimisation. It's not clear to me how this process is being carried out but as far as I can tell a single solution is being obtained for each shot. The problem is that within the measurement uncertainties and the uncertainties in the determination of A0 there will be a suite of solutions which fit the observations, and as with any inversion it is not acceptable to select only the best-fit solution. There appears to be no attempt to represent the suite of possible solutions.

Temporal variation by tidal strengthening is mentioned (Walker 2013). Could this be contributing to some of the uncertainty/range, especially at the grounding zone?

There is no mention of the free surface effect (see for example H&A2009 - for a receiver on a free surface, at normal incidence the received amplitude is double that of a receiver far from the boundary). The amplitude ratio uses $A1^2/A2$ whereas the known ratio method uses $A1/A0$ (no square). If the free surface effect is not corrected for could this cause the doubling of A0 with the multiple method as the square of A1 means this does not drop out as a ratio? Or does this fall out elsewhere?

**Specific comments**

P4L8 – Where does the -20C refer to? Floating ice? Base of firn? How is the velocity model for the sub-firn ice column determined and what is it?

P5L10 – Georod channel to channel variability greater than geophones – can you comment on why would this be?

P5eq2/P15L26 – Correct me if I am wrong but it needs to be made clear that Eq. 2 is for a basal reflection, i.e. assumed vertical through the firn. The gamma_d for diving waves referred to in Eq. 5 is in the firn and is more complicated as it must account for the ray tube energy loss (Medwin and Clay, 1998, eq 3.3.31). Presumably this is used somewhere for the direct-path pair method and should therefore be presented.

Eq3/Eq4/5 – be more explicit where equations are taken from in H&A2009. It would be helpful to label the equations with the name used to reference them in the manuscript (amplitude ratio/direct arrival etc) and perhaps set the paper structure out with similar sub-headings to make it easier to follow.

Eq. 4 – derived from H&A2009 eq5 at normal incidence – where does the factor of 2 come from on gamma_i^2?

P7L27 – What does Fig 4C,D refer to? (no labels on Fig. 4)

Table 1 – please highlight consistent columns (e.g. all means one colour, all medians another colour …). It is very difficult to read as it is presented.

P8L4 – is this method essentially using H&A2009 eq10 to determine A0? It would be useful to state this if so. As this is a new way of implementing the method I would like to see it explained with more clartity such that it can be reproduced.

P8L8 – I don't follow the argument that this is insensitive to attenuation as it is later used to calculate R (where exactly is this? Do you mean you use Zoeppritz and therefore the A0 isn't actually used?). You do a direct comparison of A0 in Table 1 which is sensitive to the choice of attenuation so it is important at that stage at least, and the result that this A0 is so different to that calculated by other methods is a significant result.

Figure 4 – Use consistent x-axis ranges as this is deceptive otherwise. I can't see ABCD labels as referred to in the text.

P10L6 – Please state the range of incidence angles at the reflector that picks are made out to. As you state later this is important in the range of forms the Zoeppritz curves will take.

P10L9 –The use of the Zoeppritz equations will require basal ice velocities and density. What values are used or are these also allowed to vary within the optimisation? It needs to made clear in the text that these are assumed/fixed and at what values (if that is the case – are Table 2 values used on grounded ice too?).

Fig. 6 – How are the uncertainties calculated and what do they represent? Why are they so much greater on the ice shelf? They are very small on the ice stream. Is this realistic given the uncertainties and range of A0?

Fig. 6/7 caption – mention that the R values use the KR method.

Fig. 6/7 and P11/L3 – The Vs uncertainties allow negative Vs velocities although the lower limit in the Zoeppritz search is zero. Vp looks to be restricted to 1440m/s although it looks like the uncertainties would take this lower given the symmetry. I suggest that negative Vs values are not plotted. This would indicate that the uncertainties are derived by error propagation which comes back to my point above about the optimisation of the inversion and accepting a single solution, the uncertainties cannot represent a suite of inversion solutions. How are negative Vs values derived by using the full A0 range with the Zoeppritz equations?

P15L10 - As you talk about transitions of 500 m it would be good to state the size of the Fresnel zone. You should then mention the scale length of the fluting and how this compares to the Fresnel zone.

P15L15-17 – I don't agree that the comparison demonstrates the efficacy of the amplitude ratio method, as stated in the following sentence, it may correlate well but it produces values twice that of the AR method. Is this not a contradiction? Or does twice the A0 value not affect estimates of R to a high degree?

---

## Author Comment (AC1) · 9 Oct 2020

**tc-2020-147 Reviewer 1 Response**

Huw Horgan et al

August 2020

The reviewer raises some useful points that highlight the need for clarification, additional work, and a wider ranging introduction and discussion. We address these below. The reviewer's comments are shown in italics and our replies are shown in plain text.

*The authors present an active seismic study in the grounding zone of the Whillans Ice Stream. Although their seismic methods are interesting, the paper needs a lot of work. Basic hypotheses (i.e., the influence of tides) about their findings are unexplored. Their clustering analysis is threadbare. Most lacking is the intellectual context. Other groups(besides folks from Penn State) do work on Whillans and it would benefit progress in the field if the authors showed more interest in interacting with these other lines of inquiry. Despite these criticisms, I do think that a significantly modified manuscript could meet publication standards in the Cryosphere.*

*Data, Methods, Results*

**1 Clustering**

*First major point: the clustering warrants more detail. How was the number of clusters chosen? It would be useful to see an L-curve analysis to persuade the reader that three is a reasonable number. Were error bounds included in the cluster analysis? It would be interesting to do so. It would also be interesting to see the clustering carried out with Vs included for the following reason. A reasonable null hypothesis would be that there exist two clusters: grounded and floating. These two clusters should be mainly distinguished by Vs. Perhaps the results indicate that grounding zone is more complicated than this simple picture? 2D scatter plots of the clustering results should be shown to demonstrate the validity of the underlying method. See https://scikit-learn.org/stable/modules/clustering.html.Different clustering methods work well with different structures in the data and it's not clear that K-Means is the right choice for this application.*

The clustering we perform is not a central part of our result but this is a good point. Our intent was to see if our results separated into floating, grounded, and

other values. This was primarily to enable reporting of representative values for the grounded and floating portions but we also hoped to highlight where the grounded portion of the ice stream exhibited properties suggestive of subglacial water. Following the advice of the reviewer we have explored this further. The reviewer's suggestion to use the scikit-learn module proved useful for this analysis. The reviewer's comment that we needed to demonstrate the suitability of the kmeans method, and the chosen number of clusters is well founded. The null hypothesis that $V_s$ should be used to discriminate makes sense, except for the fact that we are not as skilful a recovering $V_s$ as we are at recovering $R_b$, $V_p$ or $\rho$.

**1.1 Kmeans clustering**

We first present our original analysis using Kmeans and $V_p$, $R_b$ and $\rho$ (Figure 1). We have now included our uncertainties in the analysis and have seeded the initial positions with 100 random starting points. To assess the chosen number of clusters we have followed the reviewer's advice and examined the inertia values for different numbers of clusters (Figure 1A) and scatter plots of the optimal number of clusters (Figure 1C). Figure 1A indicates 3 clusters is appropriate. While the clusters clearly delineate the grounding zone (9000 m in Figure 1B) the scatter plots show that the data structures are anisotropic some clusters cuts across data structures (Figure 1C).

We then perform Kmeans clustering based on $V_s$ values as suggested. This results in the inertia curve shown in Figure 2A, which indicates three clusters is most appropriate. Clustering into two groups results in Figure 2B,C and three groups results in Figure 3. Figure 2B shows that a two cluster analysis results in clusters that are not well spatially defined and Figure 2C shows clusters cut across data structures. Three clusters (Figure 3) results in a similar mixing of clusters above and below the grounding line and clusters that cut across data structures.

Instead of using $V_s$, we suggest that clustering of normal incidence reflection coefficient $R_b$ values is appropriate. $R_b$ values should clearly distinguish water at the base of the ice, which should exhibit high amplitude negative values. $R_b$ values are also obtained independent of the non linear inversion of the Zoeppritz equations and thus provide a useful check of the methodology. The inertia plot of $R_b$ values indicates three clusters in most appropriate. We present the results of a two cluster analysis (Figure 4) and three cluster analysis (Figure 5). The two cluster analysis results in clusters almost entirely separated by the grounding line (Figure 4B). Scatter plots (Figure 4C) show the clusters do not cut across data structures. Three clusters (Figure 5) results in the floating portion of the profile separating into two cluster (Figure 5A) with two values upstream of the grounding line grouping with values from the floating portion. Scatter plots (Figure 5B) show clusters that in places cut across data structures.

We also investigated Kmeans clustering of $V_s$ and $V_p$ values (Figure 6) as high ratios of $V_p$ to $V_s$ should be a strong indicator of the presence of liquid. The inertia plot (Figure 6) indicates a two cluster analysis is most appropriate. This results in a spatial mixing of clusters above and below the grounding zone (Figure 6B), likely due to the poorly constrained $V_s$ values. Scatter plots (Figure 6C) show the clusters cut across $\rho$–$V_p$ data structures.

**1.2 DBSCAN clustering**

Due to a concern that clustering with the Kmeans algorithm may be unsuitable due to the shape and anisotropy of our clusters we investigated clustering using the DBSCAN algorithm (Ester et al., 1996). DBSCAN clusters are based on data density and the technique is relatively insensitive to outliers. DBSCAN does not require the number of clusters to be predetermined but is sensitive to a distance parameter (eps) used to define clusters and a minimum number of points used to define a cluster is also predefined. We present results with eps values determined using the nearest neighbor sorted distance method. Here we present DBSCAN clustering of $R_b$ (Figure 7), $V_s$ (Figure 8), and $V_s$ and $V_p$ (Figures 9– 10).

DBSCAN clustering of $R_b$ values is presented in Figure 7. The sorted near neighbor distance curve and the resulting number of clusters and noise points produced by a range of eps values and minimum number of samples is shown in Figure 7A. The elbow of the sorted distance curve indicates an eps value of around 0.05 is appropriate. However, we apply an eps=0.075 as values less than this result in either too many clusters or too many noise points. Even with eps=0.075 and the minimum number of points set to 4, DBSCAN results in 8 clusters and identifies 30 noise points (Figure 7B,C). These clusters group above and below the grounding zone (Figure 7B) and cut across data structures (Figure 7C).

DBSCAN clustering of $V_s$ values (Figure 8) indicates an eps setting of 0.05 is appropriate (Figure 8A). Combined with a minimum number of points set to 4, this results in 14 clusters with 6 noise points identified (Figure 8B,C). These clusters cut across the grounding line (Figure 8B) and across data structures (Figure 8C).

DBSCAN clustering of $V_s$ and $V_p$ values results in a near neighbor curve indicating an eps value of approximately 0.2 (Figure 9A). The steps in the near neighbor curve result from the variable data density. An eps set to 0.2 requires a low number of minimum points (Figure 9A), which we set to 3. This results in 7 clusters and 36 points identified as noise, with almost all the noise points located above the grounding zone (Figure 9B). Increasing the eps setting to 0.475 and increasing the minimum number of points to 4 results in just 3 clusters, which are largely separated by the grounding line (Figure 10A), and 7 noise points. Scatter plots (Figure 10B) show the clusters generally don't cut across data structures. This eps setting is, however, not justified by the near neighbor distance plot.

**1.3 Conclusion**

Both clustering approaches we have explored are not well suited to our intended purpose. Reasonable results can be obtained with either Kmeans using our previous parameters, or a Kmeans 2 cluster analysis of $R_b$. DBSCAN requires us to use higher eps values than suggested by our near neighbour analysis to obtain a useful number of clusters and noise points. The Kmeans method is likely sensitive to the anisotropy in our data, while the DBSCAN method is likely to be susceptible to the varying levels of data density. As we are primarily interested in grouping above and below the grounding zone and reporting representative values we suggest the following.

- Present the repeat kinematic GNSS analysis of the grounding zone (see next section).

- Report the mode bin values after binning of the results at an appropriate resolution.

- Archive full results with a DOI so interested parties can group or recalculate as they wish.

**2 Tidal influence**

*Second major point: Were the shots all fired at the same phase of the tides? See the work by Victor Tsai and others about tidal grounding line migration. Perhaps this is why Line 1 appears to have not hit the ocean cavity? Perhaps the ice was in contact with ocean sediments during the time of acquisition.*

**2.1 Timing of shots**

The shots were not fired at the same phase of the tide. This would not be feasible for an experiment of this scale. Figure 11 shows the timing of the shots and the tidal elevation both as a function of time, and as a function of distance along the profiles. We are unclear what the reviewer means by *Perhaps this is why Line 1 appears to have not hit the ocean cavity?* as Line 1 did hit the ocean cavity which is imaged from approximately Kilometer 9 until the end of the profile. We hope that our inclusion of Figure 12 clarifies this and we will now explicitly label the ocean cavity in the seismic profiles. Figure 11A shows that Kilometer 6–12.5 of Line 1 was acquired on the falling tide when the tidal elevation varied from +0.1 m to -0.6 m. The pronounced change in basal reflectivity that occurs at approximately Kilometer 9 on Line 1 does not coincide with a step in the tidal elevation. Other step-changes in tidal elevation along Line 1 also do not coincide with changes in basal reflectivity (e.g. Kilometer 1). Lines 2–4 all took less than a day to acquire and have no major step-changes in tidal elevation along the profiles. The onset of high basal reflectivity in Line 2 occurs in proximity to a 0.3 m change in tidal elevation offshore, however, repeat kinematic profiling (Figure 12) indicates vertical change at this location

is likely to be much less and even a 0.3 m change in water column thickness would be insufficient to cause the change in reflectivity observed.

**2.2 Tidal grounding line migration – Repeat GNSS transects across the grounding zone.**

Figure 12 shows repeat kinematic profiling along Lines 1 and 2. These repeat kinematic profiles were acquired at the times shown in the right hand panels in the figure. We have previously used these data to validate a seismically determined grounding line location (Horgan et al., 2013). We locate the grounding zone using the standard deviation in 50 m spatial bins. Upstream of the grounding zone we expect this value to represent the method uncertainty (both the GNSS observations, and our ability to repeat a track precisely) combined with a measure of the roughness of the surface. Downstream these combine with the displacement of the ice surface due to the tide. The grounding line is determined to be the point at which the standard deviation changes from values representative of grounded upstream values to those representative of floating values. The pick is subject to some interpretation as roughness varies spatially and can correlate with surface slope.

Both lines were measured using kinematic GNSS on the rising tide. The tidal range for Line 1 at the time we observed was approximately 1.5 m, while Line 2 we observed a smaller range of approximately 0.35 m. Both profiles exhibit a region of relatively-high surface slope that begins upstream of the onset of vertical tidal displacement. This zone of relatively high surface slope is observed at the grounding zone of much of the Siple and Gould Coasts (Horgan and Anandakrishnan, 2006). We pick the Line 1 grounding zone at Kilometer 9.6 for Line 1, and Kilometer 3.6 for Line 2. Well upstream of the grounding zone (>4 km) our repeat tracks typically all fall within 0.1 m vertically of each other. At the resolution of our data we do not observe migration of the grounding line in the GNSS data, nor do we observe any spatial patterns in basal reflectivity that could be attributed to tidal variation given the wavelength of our seismic source is nominally approximately 14 m in water.

**2.3 Conclusion**

In our submission we referenced earlier work that determined the grounding zone location using repeat kinematic GNSS profiling. We now see that reproducing those data here and in the manuscript would be beneficial and add clarity. We will also present the experiment timing and tide model results.

**3 Other points**

*Other points: It would be useful to also report Q values. Figure 4 (and elsewhere), What are the units of source size? What is the symbol $R_b$? The text*

*only defines $R_{bInt}$ and $R_{b10}$. Table 4. Does the plus/minus range indicate one standard deviation?*

We can present Q values for representative frequencies. The units of source size are the same as amplitude (counts). $R_b$ is normal incidence basal reflectivity and $R_{bInt}$ and $R_{b10}$ are two common ways of calculating it. The plus/minus range indicates one standard deviation.

**4 Discussion, Conclusions**

*\*\*Discussion, Conclusions\*\* I found the discussion to be quite narrow. Whillans is a complex, interesting, and unusual system. The ice stream is decelerating and is expected to be a major stabilizerof Antarctic ice loss. Yet when we go to probe the nature of the deceleration, we see that ice flow is quite bizarre and exhibits a remarkable large scale stick-slip motion. Despite this fascinating situation, the most that the authors can offer in terms of the implications for ice flow is the already-established point that "The rapid transition in basal properties indicates that the full Stokes equations are likely to be needed to besolved if the ice flow velocity field is to be accurately." The authors also mention the Walker 2013 paper. As written, this paper could be submitted to a seismology-focused journal like BSSA. As indicated in the next section of the review, it would be nicer to see closer integration with other lines of inquiry.*

*\*\*References\*\*27 out of 49 of the citations in this paper are to the Penn State group. Of the other 22citations, the average citation age is in the mid 1990's and none are about the Whillans Ice Stream. I'm surprised to see no mention of any of the numerous modeling studies that have been carried out on Whillans:*

- *Bougamont, Marion, Slawek Tulaczyk, and Ian Joughin. "Numerical investigations ofthe slow-down of Whillans Ice Stream, West Antarctica: is it shutting down like IceStream C?." Annals of Glaciology 37 (2003): 239-246.*

- *Goldberg, D., C. Schoof, and O. Sergienko (2014), Stick-slip motion of an Antarctic icestream: The effects of viscoelasticity, J. Geophy. Res. Earth Surf., 119, 1564–1580,doi:10.1002/2014JF003132.*

- *Lipovsky, B. P., and E. M. Dunham (2017), Slow-slip events on the Whillans Ice Plain,Antarctica, described using rate-and-state friction as an ice stream sliding law, J. Geo-phys. Res. Earth Surf., 122, doi:10.1002/2016JF004183.*

- *Sergienko, O. V., D. R. MacAyeal, and R. A. Bindschadler (2009), Stick–slip behavior of ice streams: Modeling investigations, Ann. Glaciol., 50(52), 87–94.*

*Similarly for the observational studies of other groups,*

- *Beem, L. H., et al. "Variable deceleration of Whillans Ice Stream, West Antarctica."C3Journal of Geophysical Research: Earth Surface 119.2 (2014): 212-224.*

- *Stearns, Leigh A., Kenneth C. Jezek, and Cornelis J. Van Der Veen. "Decadal-scale variations in ice flow along Whillans Ice Stream and its tributaries, West Antarctica."Journal of Glaciology 51.172 (2005): 147-157.*

- *Walter, J. I., E. E. Brodsky, S. Tulaczyk, S. Y. Schwartz, and R. Pettersson (2011), Transient slip events from near-field seismic and geodetic data on a glacier fault, Whillans Ice Plain, West Antarctica, J. Geophys.Res., 116, F01021,doi:10.1029/2010JF001754.*

*Other shelf/stream papers besides the Penn State Walker et al 2013 paper:*

- *Tsai, Victor C., and G. Hilmar Gudmundsson. "An improved model for tidally modulated grounding-line migration." Journal of Glaciology 61.226 (2015): 216-222.*

- *Sayag, R., and M. Grae Worster. "Elastic dynamics and tidal migration of grounding lines modify subglacial lubrication and melting." Geophysical research letters 40.22(2013): 5877-5881.*

*Finally, it is somewhat glaring that there are no citations to the WISSARD project on the Whillans Ice Stream. The authors mention basal "ponding". Other grounds call these ponds subglacial lakes:*

- *Tulaczyk, Slawek, et al. "WISSARD at Subglacial Lake Whillans, West Antarctica:scientific operations and initial observations." Annals of Glaciology 55.65 (2014): 51-58.*

- *Carter, S. P., H. A. Fricker, and M. R. Siegfried. "Evidence of rapid subglacial water piracy under Whillans Ice Stream, West Antarctica." Journal of Glaciology 59.218(2013): 1147-1162.*

- *Siegfried, Matthew R., et al. "A decade of West Antarctic subglacial lake interactions from combined ICESat and CryoSatR2 altimetry." Geophysical Research Letters C4 41.3 (2014): 891-898.*

*Note that I'm not saying the authors need to cite every one of these papers. Rather, it would simply be nice to see a little bit more interaction with the rest of the intellectual community on topics of interest.*

Our referencing reflects the methodological focus of our contribution (over-snow seismic surveying is a relatively rare method) and an ongoing body of work examining the grounding zone of Whillans Ice Stream. That said, our enthusiasm for the method has admittedly resulted in a somewhat myopic focus. We welcome the opportunity to incorporate our findings into the wider literature addressing grounding zone processes and the flow of Whillans Ice Stream. In

particular the work of Tsai and Gudmundsson, and Sayag and Worster are highly relevant and should be discussed in parallel with the study of Walker et al in both the introduction and discussion. A wider ranging discussion of the distribution of water beneath Whillans Ice Stream and associated phenomena is also appropriate.

[Figure]

Figure 1: Result of Kmeans clustering of $R_b$, $V_p$, and $\rho$ values. A) Number of Kmeans clusters (x-axis) plotted against inertia. B) Spatial distribution of seismic parameters coloured by cluster for 3 clusters. C) Scatter plots of recovered seismic parameters coloured by cluster for 3 clusters.

[Figure]

Figure 2: Result of Kmeans clustering of $V_s$ values. A) Number of Kmeans clusters (x-axis) plotted against inertia. B) Spatial distribution of seismic parameters coloured by cluster for 2 clusters. C) Scatter plots of recovered seismic parameters coloured by cluster for 2 clusters.

[Figure]

Figure 3: Result of Kmeans clustering of $V_s$ values. A) Spatial distribution of seismic parameters coloured by cluster for 3 clusters. B) Scatter plots of recovered seismic parameters coloured by cluster for 3 clusters.

[Figure]

Figure 4: Result of Kmeans clustering of $R_b$ values. A) Number of Kmeans clusters (x-axis) plotted against inertia. B) Spatial distribution of seismic parameters coloured by cluster for 2 clusters. C) Scatter plots of recovered seismic parameters coloured by cluster for 2 clusters.

[Figure]

Figure 5: Result of Kmeans clustering of $R_b$ values. A) Spatial distribution of seismic parameters coloured by cluster for 3 clusters. B) Scatter plots of recovered seismic parameters coloured by cluster for 3 clusters.

[Figure]

Figure 6: Result of Kmeans clustering of $V_s$ and $V_p$ values. A) Number of Kmeans clusters (x-axis) plotted against inertia. B) Spatial distribution of seismic parameters coloured by cluster for 2 clusters. C) Scatter plots of recovered seismic parameters coloured by cluster for 2 clusters.

[Figure]

Figure 7: DBSCAN clustering of $R_b$ values. A) Left: Sorted near neighbour distances. Right: Resulting Number of noise points and number of clusters for a range of eps and minimum sample values. B) Spatial distribution seismic properties coloured by clusters for Line 1. Clustering used eps=0.075 and a minimum number of samples = 3. Noise points shown in black. C) Scatter plots of retrieved seismic values colored by cluster.

[Figure]

Figure 8: DBSCAN clustering of $V_s$ values. A) Left: Sorted near neighbour distances. Right: Resulting Number of noise points and number of clusters for a range of eps and minimum sample values. B) Spatial distribution seismic properties coloured by clusters for Line 1. Clustering used eps=0.05 and a minimum number of samples = 4. Noise points shown in black. C) Scatter plots of retrieved seismic values colored by cluster.

[Figure]

Figure 9: DBSCAN clustering of $V_s$ and $V_p$ values. A) Left: Sorted near neighbour distances. Right: Resulting Number of noise points and number of clusters for a range of eps and minimum sample values. B) Spatial distribution seismic properties coloured by clusters for Line 1. Clustering used eps=0.2 and a minimum number of samples = 3. Noise points shown in black. C) Scatter plots of retrieved seismic values colored by cluster.

[Figure]

Figure 10: DBSCAN clustering of $V_s$ and $V_p$ values. A) Spatial distribution seismic properties coloured by clusters for Line 1. Clustering used eps=0.475 and a minimum number of samples = 4. Noise points shown in black. B) Scatter plots of retrieved seismic values colored by cluster.

[Figure]

Figure 11: Shot timing and vertical tidal elevation. A) Line 1. Top subplot shows the timing of shots (blue bars) overlain on the vertical tidal elevation from Tide Model Driver (Pandman and Erofeeva (2005). Bottom subplot shows the Tide Model Driver (Padman and Erofeeva, 2005) elevation at the time of shooting as a function of distance along the profile. B-C) same as A) but for lines 2, 3, and 4, respectively.

[Figure]

Figure 12: Repeat kinematic profiling along Lines 1 (A,B) and 2 (C,D). Panels A,C show the elevation (top), residual elevation after removal of a best-fitting spline (middle), and standard deviation of residual elevation in 50 m spatial bins (bottom). Panels B,D show the timing of the GNSS profiles overlain on vertical elevation component from Padman and Erofeeva's (2005) Tide Model Driver.

---

## Author Comment (AC2) · 9 Oct 2020

**tc-2020-147 Reviewer 2 Response**

Huw Horgan et al

August 2020

**1 Introduction**

*Review of "Grounding zone subglacial properties from calibrated active source seismic methods" by H. Horgan et al Alex Brisbourne, August 2020.*

*The authors present an assessment of active seismic data analysis methods using measurements made at the grounding zone of Whillans Ice Stream. Data cover both grounded and floating ice and therefore present an opportunity to assess and calibrate existing and new data processing methods used to obtain absolute properties of the subsurface. To this end the paper is a useful addition to studies of this nature and builds upon the previous work of Holland and Anandakrishnan (2009) (from here referred to as HA2009). The manuscript is well written and structured. However, as outlined in my comments below there are a number of clarifications needed in order that the reader can ascertain exactly how the analysis is applied and how closely this fits with previous work. The methodology description in insufficient in places and clearer self-referencing would improve the readers' ability to follow the methodology.*

*General comments Section 2.6 - Estimating subglacial properties – Optimisation. It's not clear to me how this process is being carried out but as far as I can tell a single solution is being obtained for each shot. The problem is that within the measurement uncertainties and the uncertainties in the determination of A0 there will be a suite of solutions which fit the observations, and as with any inversion it is not acceptable to select only the best-fit solution. There appears to be no attempt to represent the suite of possible solutions.*

In our submission we attempted to represent possible solutions in the following way:

- We assume the dominant source of uncertainty is from our estimate of source size.

- We approximate the uncertainty in our source size using the standard deviation of the source size distribution (Figure 4, P8 L15-16.)

- We estimate normal incidence basal reflectivity ($R_{bInt}$, $R_{b10}$) using the average value and the average $\pm$ one standard deviation of the source size. No inversion is required at this stage. We estimate a solution for each shot. No mixing, smoothing, or spatial normalisation is applied.

- To obtain more information about the substrate we invert the reflection amplitude versus offset for seismic properties ($V_p$, $V_s$, $\rho$) using the average source size, and the average source size $\pm$ one standard deviation of the source size (P11,1-2). These source sizes are all propagated through the inversion to retrieve seismic properties. We do this for each shot with no mixing, smoothing, or spatial normalisation.

- We plot the result for the mean source size with symmetrical error bars. The size of the error bars are the maximum difference from the mean retrieved seismic properties and those retrieved using the low ($A_0$-$1\sigma$) and high estimates of source size ($A_0$+$1\sigma$).

We agree that plotting negative velocities or densities is not helpful. We are also interested in better quantifying uncertainties through the inversion. Our existing method can produce very small uncertainties as inversion from a range of source sizes can result in a very narrow band of results, or the same result. This is exacerbated by our use of constrained nonlinear methods. One alternate approach would be instead of limiting ourselves to the mean and standard deviation to instead provide an ensemble of source sizes based on the observed distribution of sources and present the ensemble of results. Where this still results in too narrow a range of uncertainty a minimum value could be set, informed by our $R_b$ results.

*Temporal variation by tidal strengthening is mentioned (Walker 2013). Could this be contributing to some of the uncertainty/range, especially at the grounding zone?*

[Walker and others, 2013] suggest a long-term temporal strengthening upstream of the grounding zone in their fixed-fulcrum model. In their simulations this strengthening would reach a maximum approximately 1.2 km upstream of the grounding zone for ice that is 1 km thick. While our results show overall stiffer till than that observed elsewhere beneath the Siple Coast ice streams, they do not show a clear pattern in stiffness. If the reviewer is suggesting the possibility of variable results at different stages of the tidal cycle we refer them to our response to Reviewer 1's comments where we explore the relationship between our results and the stage of the tide.

*There is no mention of the free surface effect (see for example HA2009 - for a receiver on a free surface, at normal incidence the received amplitude is double that of a receiver far from the boundary). The amplitude ratio uses A12/A2 whereas the known ratio method uses A1/A0 (no square). If the free surface effect is not corrected for could this cause the doubling of A0 with the multiple*

*method as the square of A1 means this does not drop out as a ratio? Or does this fall out elsewhere?*

The free amplitude scalar referred to by [Holland and Anandakrishnan, 2009] applies to normal incidence rays in an isotropic medium with the receiver at the free surface. We currently account for this effect using the amplification approximation provided by [Shearer, 2009][Equation 6.19] that uses the square root of the impedance contrast between the source and receiver locations. Our geophones are buried at approximately 0.5 m depth and our shots are at approximately 27 m depth resulting in a correction to our path amplitude factors of approximately $\sqrt{10}$. A more complete treatment is provided by [Aki and Richards, 1980], and a comparison between the [Shearer, 2009] approximation and the [Aki and Richards, 1980] approach are planned as future work.

*Specific comments P4L8 – Where does the -20C refer to? Floating ice? Base of firn? How is the velocity model for the sub-firn ice column determined and what is it?*

-20C was chosen as a representative temperature for the ice column (see [Paterson, 1994] Fig. 10.6). Below the depth constrained by our shallow refraction model, our velocity model consists of a linear extrapolation to a $V_p$ corresponding to -20C (3860 m s$^{-1}$). We keep this velocity constant to the base of the ice. [Kohnen, 1974] demonstrate an increase in $V_p$ of 2.3 m s$^{-1}$ per degree C, so we are fairly insensitive to our choice of temperature. Out ray tracing is however sensitive to the temperature structure and inversions in the temperature model would lead to diving rays that complicate the modelling of direct arrivals. This will require more investigation if ray tube effects for direct arrivals are to be accurately estimated.

*P5L10 – Georod channel to channel variability greater than geophones – can you comment on why would this be?*

We suspect the variability we observe results from less consistent coupling of the georods to the surrounding snow and firn. Our method was to bury the georod in a shallow (approximately 0.5 m deep) trench. (We used georods where the components were vertical when the unit was placed horizontally.) The geophones had short spikes that were inserted into the snow/firn, preferably into a hard layer, at a depth of approximately 0.5 m.

*P5eq2/P15L26 – Correct me if I am wrong but it needs to be made clear that Eq. 2 is for a basal reflection, i.e. assumed vertical through the firn. The $\gamma_d$ for diving waves referred to in Eq. 5 is in the firn and is more complicated as it must account for the ray tube energy loss (Medwin and Clay, 1998, eq 3.3.31). Presumably this is used somewhere for the direct-path pair method and should therefore be presented.*

This is a good point. We ran our analysis for the Known Reflector method, and Primary–Multiple Ratio method with both the $\gamma$ estimate we present in our submission and a more complete ray tube treatment. The results are largely similar and we chose to present the simpler 1/distance $\gamma$ estimate for simplicity. The more complete treatment, which uses the square root of wavefront energy, does however result in a linear regression gradient between the two methods that is closer to 1:1 (1.6 versus 2.0). We have not run the direct–path methods with a ray tube approach and will endeavor to do so.

*Eq3/Eq4/5 – be more explicit where equations are taken from in HA2009. It would be helpful to label the equations with the name used to reference them in the manuscript (amplitude ratio/direct arrival etc) and perhaps set the paper structure out with similar sub-headings to make it easier to follow.*

Will do. We have changed some subscripts but will make it clear which equations are from [Holland and Anandakrishnan, 2009]. We will also label the equations with the names we use to refer to them, and look to make the structure similar to [Holland and Anandakrishnan, 2009] where possible. Thanks for these suggestions.

*Eq. 4 – derived from HA2009 eq5 at normal incidence – where does the factor of 2 come from on $\gamma_i^2$?*

This is a typo. The correct equation was implemented in our analysis. Thank you for pointing this out.

*P7L27 – What does Fig 4C,D refer to? (no labels on Fig. 4)*

These refer to an earlier draft. Apologies for the confusion.

*Table 1 – please highlight consistent columns (e.g. all means one colour, all medians another colour ...). It is very difficult to read as it is presented.*

We will follow this advice.

*P8L4 – is this method essentially using HA2009 eq10 to determine A0? It would be useful to state this if so. As this is a new way of implementing the method I would like to see it explained with more clartity such that it can be reproduced.*

Yes, we use [Holland and Anandakrishnan, 2009] equation 10 with the source amplitude obtained using the known reflector method. Our method is an optimisation of [Holland and Anandakrishnan, 2009] equation 10 where we minimise the misfit between our $R(\theta)$ estimated using our $A_0$ and the $R_\theta$ resulting form a range of possible acoustic properties.

*P8L8 – I don't follow the argument that this is insensitive to attenuation as it is later used to calculate R (where exactly is this? Do you mean you use Zoeppritz and therefore the A0 isn't actually used?).*

Our method of estimating source amplitude using a known reflector requires an estimate of attenuation. Our method of estimating basal reflectivity also requires an estimate of attenuation. Both these steps use [Holland and Anandakrishnan, 2009] Equation 10. As the same attenuation is used in both steps the resulting $R(\theta)$ is independent of the attenuation chosen.

*You do a direct comparison of A0 in Table 1 which is sensitive to the choice of attenuation so it is important at that stage at least, and the result that this A0 is so different to that calculated by other methods is a significant result.*

That is correct. Changing the attenuation does change the value of $A_0$, which is why we use the strength of the regression to assess the relationship between methods not the gradient. We chose to use reasonable values for attenuation instead of tweaking our attenuation to force a 1:1 gradient when comparing our $A_0$ estimates.

*Figure 4 – Use consistent x-axis ranges as this is deceptive otherwise. I can't see ABCD labels as referred to in the text.*

We will make these changes.

*P10L6 – Please state the range of incidence angles at the reflector that picks are made out to. As you state later this is important in the range of forms the Zoeppritz curves will take.*

We will quantify this for each line. It is dependent on ice thickness due to interference from the direct arrivals. Almost all shots have reflector picks out to 25 degrees with some having picks out to 30 degrees.

*P10L9 –The use of the Zoeppritz equations will require basal ice velocities and density. What values are used or are these also allowed to vary within the optimisation? It needs to made clear in the text that these are assumed/fixed and at what values (if that is the case – are Table 2 values used on grounded ice too?).*

Ice properties are set to the values in Table 2 for both grounded and floating ice. These were not allowed to vary in the inversion. The sensitivity of our results to these values could be done either by allowing them to vary in a constrained way in the inversion, or by forward modelling possible values.

*Fig. 6 – How are the uncertainties calculated and what do they represent? Why are they so much greater on the ice shelf? They are very small on the ice stream. Is this realistic given the uncertainties and range of A0?*

Please see our earlier comments regarding uncertainty estimation. The small uncertainties result from $A_0 \pm 1\sigma$ resulting in the same inversion result. Adopted the uncertainty analysis we suggest above, including a minimum value should result in more representative estimates.

*Fig. 6/7 caption – mention that the R values use the KR method.*

Will do.

*Fig. 6/7 and P11/L3 – The Vs uncertainties allow negative Vs velocities although the lower limit in the Zoeppritz search is zero. Vp looks to be restricted to 1440m/s although it looks like the uncertainties would take this lower given the symmetry. I suggest that negative Vs values are not plotted. This would indicate that the uncertainties are derived by error propagation which comes back to my point above about the optimisation of the inversion and accepting a single solution, the uncertainties cannot represent a suite of inversion solutions. How are negative Vs values derived by using the full A0 range with the Zoeppritz equations?*

Please see our earlier comments regarding uncertainty estimation.

*P15L10 - As you talk about transitions of 500 m it would be good to state the size of the Fresnel zone. You should then mention the scale length of the fluting and how this compares to the Fresnel zone.*
The width of our first Fresnel zone is approximately 240 m (100 Hz signal at 760 m depth in a 3860 m s$^{-1}$ medium) and the corresponding quarter wavelength is approximately 9.7 m. The fluting modelled by [Christianson and others, 2016] has a wavelength of 20 m; amplitudes of 5.75 m; and RMS heights of 4 m.

*P15L15-17 – I don't agree that the comparison demonstrates the efficacy of the amplitude ratio method, as stated in the following sentence, it may correlate well but it produces values twice that of the AR method. Is this not a contradiction? Or does twice the A0 value not affect estimates of R to a high degree?*

As we state earlier, we can reduce the gradient of the relationship by varying our attenuation estimate and path amplitude factor ($\gamma$). We have chosen not to do this as we think using values widely adopted in glaciology better emphasises areas where both methods are deficient. If we adjust our attenuation values until we produce a good fit between our Known Reflector method and the Primary-Multiple Ratio method the result would be misleadingly good. However, the methods do correlate well, as apposed to our analysis of direct path methods. We will clarify our language and thinking in this regard.

In closing we thank the reviewer for their detailed and constructive review. We appreciate the time and thought that went into it.

**References**

[Aki and Richards, 1980] Aki, Keiiti and Paul G. Richards, 1980. Quantitive Seismology. Theory and Methods, W.H. Freeman and Co.

[Christianson and others, 2016] Christianson, Knut, Robert W. Jacobel, Huw J. Horgan, Richard B. Alley, Sridhar Anandakrishnan, David M. Holland and Kevin J. DallaSanta, 2016. Basal conditions at the grounding zone of Whillans Ice Stream, West Antarctica, from ice-penetrating radar, *Journal of Geophysical Research: Earth Surface*, **121**(11), 1954–1983.

[Holland and Anandakrishnan, 2009] Holland, C. and S. Anandakrishnan, 2009. Subglacial seismic reflection strategies when source amplitude and medium attenuation are poorly known, *Journal of Glaciology*, **55**(193), 931–937.

[Kohnen, 1974] Kohnen, H., 1974. The temperature dependence of seismic waves in ice, *Journal of Glaciology*, **13**(67), 144–147.

[Paterson, 1994] Paterson, W. S. B., 1994. The Physics of Glaciers, Pergamon, Tarrytown, N. Y., 3rd ed.

[Shearer, 2009] Shearer, Peter M., 2009. Introduction to seismology, Cambridge University Press.

[Walker and others, 2013] Walker, R. T., B. R. Parizek, R. B. Alley, S. Anandakrishnan, K. L. Riverman and K. Christinason, 2013. Ice-shelf tidal flexure and subglacial pressure variations, *Earth and Planetary Science Letters*, **361**, 422–428.

---

## Referee Report (RR1)

**Comments on revised manuscript (version4) "Grounding zone subglacial properties from calibrated active source seismic methods" by H. Horgan et al**

**Alex Brisbourne, January 2021**

The authors have made significant changes to the original manuscript. This has improved the clarity and completeness of the paper. The authors have in general addressed the points raised in my original review with one omission as detailed below. The removal of the clustering (in response to Reviewer #1) has, to my mind, improved the manuscript by removing unnecessary distractions from the key messages. There are some important results pertaining to reliable data analysis for basal properties.

There is still an issue with regards the free surface effect, as discussed on Page 3 of Reviewer 2 response. Shearer (2009) Eq 6.19 accounts for the amplification effect due to conservation of energy that results from the density contrast between the source and receiver depths. However, this amplification is in addition to the free surface effect which is a doubling of amplitude at the free surface due to the conservation of energy (which I would argue is still pertinent with a receiver burial depth of 0.5 m). It just so happens that all these methods rely on amplitude ratios to derive A0 and the free surface effect therefore drops out. However, for completeness it should be included, or words to this effect included, as per H&A2009.

**Additional minor comments**

**P8L23** (H&A2009, Eq. 9)

**P9L18** should this be $A_0$ rather than $R(\theta)$

**P17 Figure 8** – for clarity plot the red tidal signal on top of the blue bars

**P18 L3** – reproduce not reproduced

---

## Author Response (AR2)

ANTARCTIC RESEARCH CENTRE TE PUTAHI RANGAHAU I TE KOPAKATANGA KI TE TONGA VICTORIA UNIVERSITY OF WELLINGTON, PO Box 600, Wellington 6140, New Zealand Phone + 64-4-463 6587 Fax + 64-4-463 5186 Email antarctic-research@vuw.ac.nz Web www.victoria.ac.nz/antarctica

**Re: tc-2020-147 Resubmission. Response to Reviewer 2 and Editor Comments.**

We thank the editor and reviewer for their additional comments. In the following we reproduce these comments and suggestions in *italics* with our replies shown in plain text.

**Editors Comments**

Please respond to the minor concerns raised by RV-2 and incorporate the suggested editorial remarks. Please also make sure that all Figures are placed as intended. RV-1 has mentioned possibly missing Figures that appear in the response-to-reviews but did not make it into the final version (I think they did, but just make sure that this is indeed the case).

We have reviewed the figures and they all appear present and in the correct place. The reviewer may have been referring to the clustering analysis which was included in the reply to the reviewer but was deliberately not included in the resubmission. This was discussed in the reply to reviewer and further supported by Reviewer 2.

Minor edits

Eq (6) appears misplaced (should be close to l. 17 on p. 9). Change made.

Table 2 add units. Change made.

Table 3 add units. Change made.

**Reviewer 2 Comments**

The authors have made significant changes to the original manuscript. This has improved the clarity and completeness of the paper. The authors have in general addressed the points raised in my original review with one omission as detailed below. The removal of the clustering (in response to Reviewer 1) has, to my mind, improved the manuscript by removing unnecessary distractions from the key messages. There are some important results pertaining to reliable data analysis for basal properties.

ANTARCTIC RESEARCH CENTRE TE PUTAHI RANGAHAU I TE KOPAKATANGA KI TE TONGA VICTORIA UNIVERSITY OF WELLINGTON, PO Box 600, Wellington 6140, New Zealand Phone + 64-4-463 6587 Fax + 64-4-463 5186 Email antarctic-research@vuw.ac.nz Web www.victoria.ac.nz/antarctica

There is still an issue with regards the free surface effect, as discussed on Page 3 of Reviewer 2 response. Shearer (2009) Eq 6.19 accounts for the amplification effect due to conservation of energy that results from the density contrast between the source and receiver depths. However, this amplification is in addition to the free surface effect which is a doubling of amplitude at the free surface due to the conservation of energy (which I would argue is still pertinent with a receiver burial depth of 0.5 m). It just so happens that all these methods rely on amplitude ratios to derive A0 and the free surface effect therefore drops out. However, for completeness it should be included, or words to this effect included, as per HA2009.

We appreciate the reviewer persisting with this comment. We have now included additional text addressing this. This additional text reads:

"H&A2009 noted that placing receivers at a free surface results in a further doubling of recorded amplitudes for normal incidence returns. We tested including free surface amplification but did not apply it to the analysis presented here due to the burial of our receivers, although the shallow burial depth of 0.5 m may justify its inclusion. If included, this additional amplification would have resulted in a halving of the source sizes for two of our methods (the multiple bounce method and the known reflector method (Sections 2.4.1 and 2.4.3, Table 1). Including free surface amplification would have had a small effect (

ANTARCTIC RESEARCH CENTRE TE PUTAHI RANGAHAU I TE KOPAKATANGA KI TE TONGA VICTORIA UNIVERSITY OF WELLINGTON, PO Box 600, Wellington 6140, New Zealand Phone + 64-4-463 6587 Fax + 64-4-463 5186 Email antarctic-research@vuw.ac.nz Web www.victoria.ac.nz/antarctica

In closing we again thank the reviewer and editor for their constructive reviews. We appreciate the time and thought that went into them.

Sincerely

An for

Dr. Huw J. Horgan Associate Professor Antarctic Research Centre School of Geography Environment and Earth Sciences Victoria University of Wellington